# A deep learning approach for the analysis of birdsong

**Therese MI Koch\*, Ethan S Marks, Todd F Roberts\***

Department of Neuroscience, UT Southwestern Medical Center, Dallas, United States

## eLife Assessment

This work introduces a new Python package, Avian Vocalization Analysis (AVN) that provides several key analysis pipelines for birdsong research. This tool is likely to prove **useful** to researchers in neuroscience and beyond, as demonstrated by **convincing** experiments using a wide range of publicly available birdsong data.

**\*For correspondence:**
therese.koch1@gmail.com
(TMIK);
todd.roberts@utsouthwestern.
edu (TFR)

**Competing interest:** The authors declare that no competing interests exist.

## Abstract

Deep learning tools for behavior analysis have enabled important insights and discoveries in neuroscience. Yet, they often compromise interpretability and generalizability for performance, making quantitative comparisons across datasets difficult. We developed a novel deep learning-based behavior analysis pipeline, Avian Vocalization Network (AVN), for zebra finch learned vocalizations - the most widely studied vocal learning model species. AVN annotates songs with high accuracy, generalizing across multiple animal colonies without re-training, and generates a comprehensive set of interpretable features describing song syntax, timing, and acoustic properties. We use this feature set to compare song phenotypes across research groups and experiments and to predict a bird's stage in song development. Additionally, we have developed a novel method to measure song imitation that requires no training data for new comparisons and outperforms existing similarity scoring methods in its sensitivity and agreement with expert human judgements. These tools are available through the open-source AVN python package and graphical application, making them accessible to researchers without prior coding experience. Altogether, this behavior analysis toolkit stands to accelerate the study of vocal behavior by standardizing phenotype and learning outcome mapping, thus helping scientists better link behavior to the underlying neural processes.

## Introduction

A deep understanding of animal behavior is fundamental to a deep understanding of the brain. However, an accurate, quantitative description of animal behavior, particularly in ethologically relevant contexts, remains a substantial challenge in neuroscience research. In recent years, careful observation of motor and vocal behaviors is increasingly being replaced with machine learning and deep-learning-based approaches. These tools allow researchers to consider much greater volumes of data than was previously possible and uncover patterns in animal behavior that are undetectable to humans and have led to important insights into ethologically relevant behaviors and the effects of experimental interventions thereupon (*Wiltschko et al., 2020*; *Hsu and Yttri, 2021*; *Alam et al., 2024*). However, this increased power often comes at the expense of interpretability and generalizability.

An increasing number of supervised deep learning methods are being developed for the automated annotation of animal vocalization behavior (*Steinfath et al., 2021*; *Cohen et al., 2022*; *Gu et al., 2023*; *Coffey et al., 2019*), and unsupervised methods for dimensionality reduction and analysis (*Alam et al., 2024*; *Goffinet et al., 2021*; *Brudner et al., 2023*; *Sainburg et al., 2020*; *Roeser, 2023*). While unsupervised approaches are very powerful and have been shown to explain more

variance in vocalization repertoires than hand-selected acoustic features (*Goffinet et al., 2021*), the features that they generate are notoriously difficult to interpret and specific to the exact dataset from which they were derived (*Alam et al., 2024*; *Goffinet et al., 2021*; *Brudner et al., 2023*). As a result, unsupervised data-driven methods, while allowing detailed comparison of individuals within the same data set, make it more difficult to compare the nature and severity of vocal phenotypes across experiments and research groups.

To truly maximize the benefits of machine learning and deep learning methods for behavior analysis, their power must be balanced with interpretability and generalizability. This can be achieved by combining automated annotation with a carefully selected set of meaningful features, thereby creating a common feature space for the comparison of behavioral phenotypes across research groups, experimental conditions, and studies. For speed, ease of use, and standardization, the annotations should be generated without the need for any training data or hyperparameter setting for new individuals or recording conditions. The features should be consistent across recording conditions, allowing direct, meaningful comparisons between research groups. The feature set should be comprehensive, describing multiple aspects of the behavior. Finally, the features should be interpretable, allowing researchers to form concrete hypotheses about how different manipulations will affect specific features and use observed feature values to guide future experimental design.

We have developed an analysis pipeline called *Avian Vocalization Network (AVN)* which satisfies these criteria for zebra finch song analysis. Zebra finches are the most popular animal model for the study of vocal learning. They learn to sing a single, highly stereotyped song by memorizing the song of an adult tutor, then refining their vocalizations to match this song template during a sensorimotor learning period early in development (*Figure 1a*); a process which bears many parallels to human speech learning (*Doupe and Kuhl, 1999*). Typical zebra finch songs consist of a variable number of introductory notes, followed by multiple repetitions of a motif, composed of 3–10 unique syllable types produced in a stereotyped sequence. Traditionally, zebra finch song has been analyzed by segmenting the song into syllables, then manually labeling syllables based on visual inspection of their spectrograms (*Lachlan et al., 2016*; *Tchernichovski and Nottebohm, 1998*; *Scharff and Nottebohm, 1991*). This process is very labor-intensive, which limits the number of songs that can be considered at a time. Manual syllable labeling and motif identification can also be subjective and therefore susceptible to experimenter bias, as motif composition and syllable types can be somewhat ambiguous, particularly in young birds with immature song and in birds with experimentally disrupted songs.

We tested and compared two deep-learning approaches for syllable segmentation, which don't require any additional training data or hyperparameter setting for new birds. These were WhisperSeg (*Gu et al., 2024*) and a novel application of Tweetynet (*Cohen et al., 2022*). We then applied unsupervised dimensionality reduction and clustering methods to assign labels to these automatically segmented syllables, as in *Sainburg et al., 2020*, and used thorough validation approaches to examine how these unsupervised methods perform when combined with the segmentation approaches. Finally, we use the resulting annotated songs to calculate a set of 55 interpretable features which describe the syntax, timing, and acoustic properties of a set of songs (*Figure 1b*). We show that the automated annotation performs consistently well across multiple zebra finch colonies, and that the feature set can be used to glean mechanistic insights from the comparison of vocal phenotypes, and to predict a bird's stage in song development (*Figure 1c*). We also developed a new method to compare two birds' syllable repertoires to measure song learning, which outperforms existing song similarity scoring methods on multiple key metrics. The complete pipeline is available as an open-source Python package and as an application with a graphical user interface, allowing researchers with no prior coding experience to easily annotate their songs, calculate the feature set, and calculate song similarity scores (*Figure 2—figure supplement 1*).

## Results

### Comparing deep learning methods for fully automated syllable segmentation

To accurately segment and label zebra finch songs without the need for any individual-specific training data or hyperparameter tuning, we tested and compared two different deep-learning-based approaches for syllable segmentation. Traditionally, zebra finch song is segmented based on an

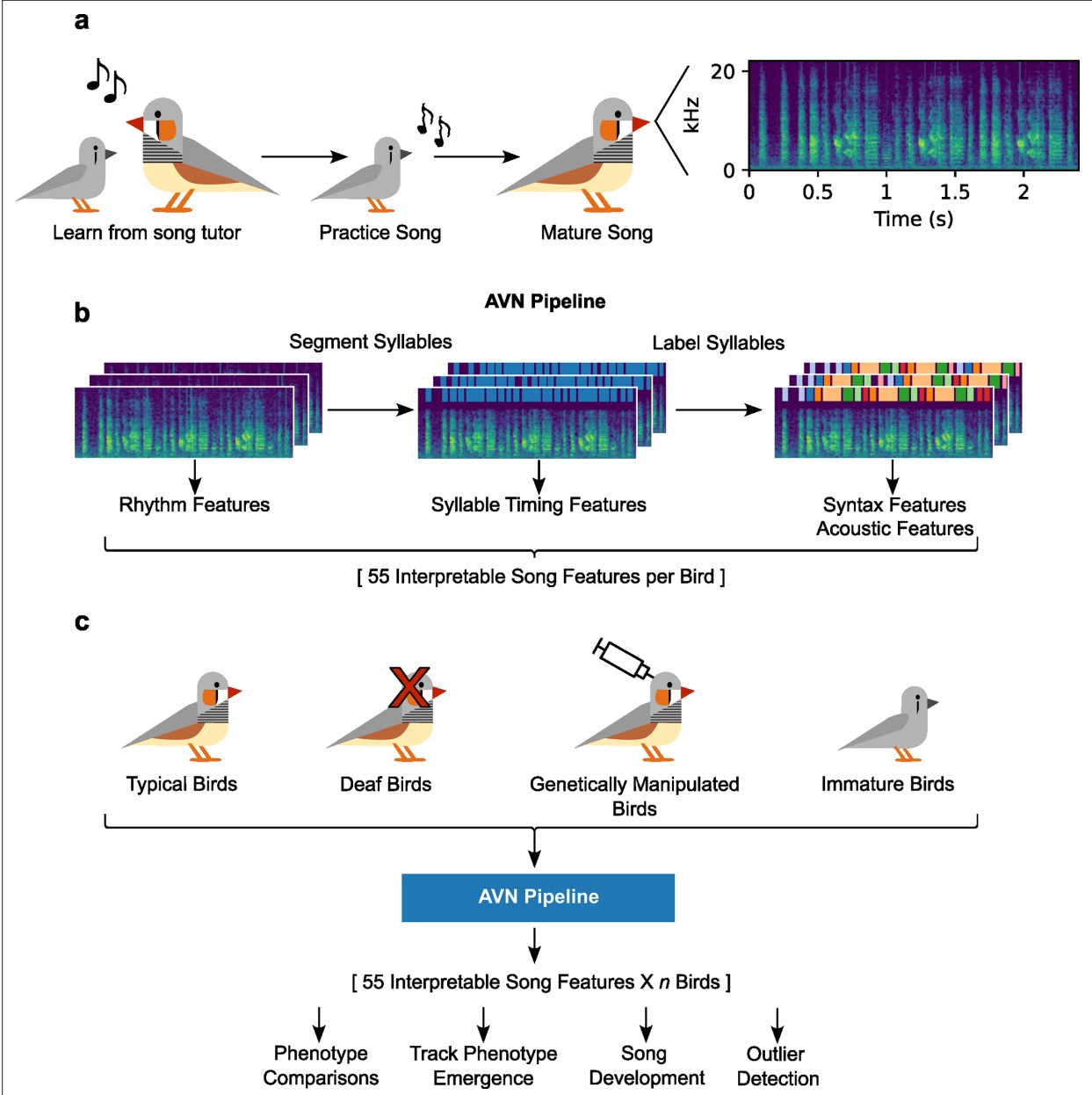

**Figure 1.** Overview of AVN song analysis pipeline. (**a**) Schematic timeline of zebra finch song learning. (**b**) Overview of AVN song analysis pipeline. Spectrograms of songs are automatically segmented into syllables, then syllables are labeled. The raw spectrograms are used to calculate features describing the rhythm of a bird's song, the segmentations are used to calculate syllable-level timing features, and the labeled syllables are used to calculate syntax-related features and acoustic features of a bird's song. (**c**). Birds from different research groups, with multiple different song phenotypes can all be processed by the AVN pipeline, generating a matrix of directly comparable, interpretable features, which can be used for downstream analyses including phenotype comparisons, tracking the emergence of a phenotype over time, investigating song development, and detecting individual outlier birds with atypical song phenotypes.

The online version of this article includes the following figure supplement(s) for figure 1:

**Figure supplement 1.** AVN graphical application examples.

amplitude threshold (***Sainburg et al., 2020***; ***Koumura and Okanoya, 2016***). The best value for this amplitude threshold depends heavily on recording conditions and background noise levels, and setting this threshold often requires careful trial and error by a human annotator. Amplitude-based segmentation methods also cannot distinguish between song syllables and noises, like wing flaps or other non-vocal artifacts, which can contaminate downstream analyses. Instead of relying on amplitude alone,

we compared two deep learning models, TweetyNet (*Cohen et al., 2022*) and WhisperSeg (*Gu et al., 2024*), which take the full spectral content of the audio into account when performing segmentation.

We tested these two segmentation methods with a dataset of over 1,000 manually annotated songs from 35 adult zebra finches, including birds with typical song production, isolate birds raised without a song tutor, and birds with disrupted song production due to knockdown of the transcription factor FoxP1 (FP1) in the song nucleus HVC (*Garcia-Oscos et al., 2021*). The annotation consists of manual correction of amplitude-segmented syllables and assignment of syllable labels based on the appearance of the spectrograms by expert annotators. TweetyNet was designed to simultaneously segment and label syllable types by assigning syllable labels to short spectrogram frames. This application requires re-training for each individual bird, so we instead trained TweetyNet to label spectrogram frames as simply containing vocalizations, silence, or noise. We trained it with 34 of the 35 birds in the dataset and evaluated segmentation accuracy with the remaining bird, repeating this once for each bird in the dataset. This novel approach allows the model to learn an abstract notion of vocalization vs. non-vocalization that generalizes well to new individuals not included in training. The WhisperSeg model is already trained for segmentation of new individuals, so we used the existing standard model to segment each of our birds. Segmentation accuracy was evaluated against expert human annotations by calculating the precision, recall, and F1 scores of syllable onset detections within 10 ms of a syllable onset in the manual annotations.

WhisperSeg shows the best performance, with a mean F1 score of 0.882 (± SEM 0.02), compared to TweetyNet's score of 0.824 (± SEM 0.03) and a simple amplitude segmentation algorithm (RMSE) with a mean score of 0.593 (± SEM 0.02; *Figure 2a*). WhisperSeg's precise onset times were also more consistent with expert human annotations than both other methods (median absolute time difference of 1.75 ms for WhisperSeg, 2.22 ms for TweetyNet, and 3.81 ms for RMSE; *Figure 2b*). All three methods performed similarly for the typical, isolate, and FP1 KD birds (*Figure 2—figure supplement 1a–f*). As a further test of the generalization of these methods, we applied them to a dataset of manually annotated songs from 25 birds from the Rockefeller University Field Research Center Song Library (*Tchernichovski et al., 2021*). Using the pre-trained WhisperSeg model and a TweetyNet model trained on all 35 birds from the UTSW colony and none from the Rockefeller Song Library, each of these models yielded very similar segmentation accuracy scores to those obtained with the UTSW colony (*Figure 2a and c*; *Figure 2—figure supplements 1–2*).

## Accurate, fully unsupervised syllable labeling

Next, we assign syllable labels to these segmented units. To achieve this, we first performed UMAP dimensionality reduction (*McInnes et al., 2018*) on spectrograms of the segmented syllables, then performed hierarchical density-based clustering (HDBSCAN; *McInnes and Healy, 2017*) on the syllables' UMAP embeddings, as in *Sainburg et al., 2020*. We calculated the UMAP embeddings of all segmented syllables for each of the 35 birds in our dataset using manual segmentations, WhisperSeg, or TweetyNet segmentations. In all cases, we found that the syllables formed multiple dense clusters, which corresponded well to manually annotated syllable labels (*Figure 2d–g*; *Figure 2—figure supplement 3*).

Using manual segmentation yielded the best agreement with manual labels (mean v-measure score = 0.87 +- 0.01), which is to be expected, as no discrepancies are introduced during segmentation (*Figure 2g*). When clustered, WhisperSeg's segments yielded better agreement with manual labels than TweetyNet's (WhisperSeg mean v-measure = 0.80 +- 0.02, TweetyNet's mean v-measure = 0.77 +– 0.02). We observe similar performance on our second dataset of 25 typical adult zebra finches from the Rockefeller Song Library, suggesting that these methods generalize well across colonies and recording environments (*Figure 2g*). UMAP embeddings have an inherent stochasticity, meaning that running this embedding and clustering multiple times on the same dataset will result in slight changes to the embeddings and labels across runs. To measure the degree of these changes, we calculated the v-measure score for clustering labels for each of 30 different random initializations of the UMAP embedding for each of 25 birds from the UTSW colony and found negligible variation in the v-measure score across runs (*Figure 2—figure supplement 4*).

Most of the errors in TweetyNet's syllable labeling stem from inconsistencies in syllable segmentation. For example, if two syllables with labels 'a' and 'b' in the manual annotation are segmented sometimes as separate syllables and sometimes as a merged syllable 'ab', the clustering is likely to

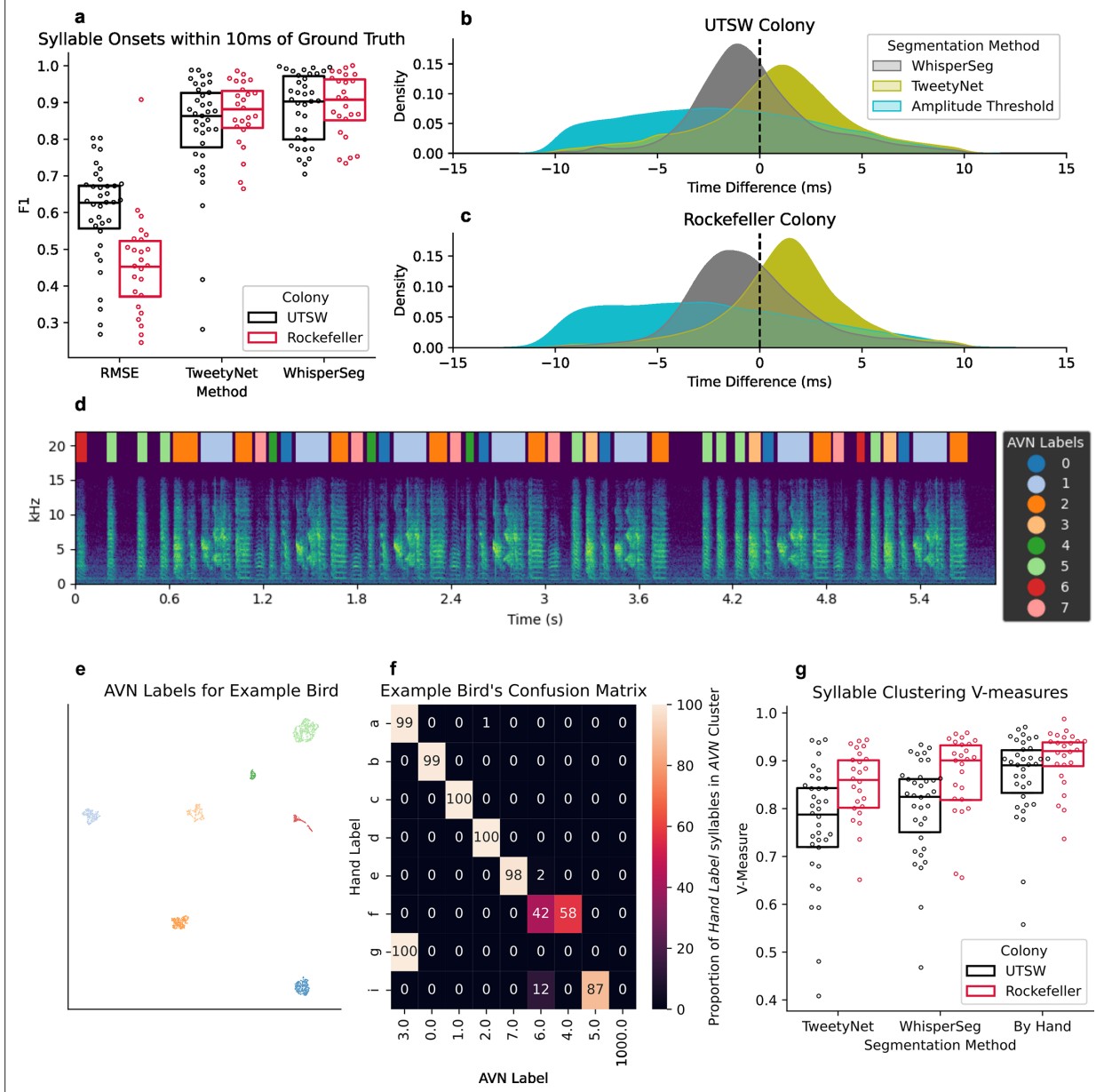

**Figure 2.** Automated syllable annotation metrics. (**a**) F1 scores for syllable onset detections within 10 ms of a syllable onset in the manual annotations of each bird (n=35 from UTSW and n=25 from Rockefeller) across segmentation methods. (**b**) Distribution of time differences between predicted syllable onsets and their best matches in the manual annotation, across segmentation methods. Distributions include all matched syllables across all 35 birds from the UT Southwestern colony (UTSW) and (**c**) 25 from Rockefeller. (**d**) Example spectrogram of a typical adult zebra finch. The song was segmented with WhisperSeg and labeled using UMAP and HDBSCAN clustering. Colored rectangles reflect the labels of each syllable. (**e**) Example UMAP plot of 3131 syllables from the same bird as in **d** and **f**. Each point represents one syllable segmented with WhisperSeg, and colors reflect the AVN label of each syllable. (**f**) Example confusion matrix for the bird depicted in **d** and **e**. The matrix shows the percentage of syllables bearing each manual annotation label which fall into each of the possible AVN labels. (**g**) V-measure scores for AVN syllable labels compared to manual annotations for each bird (n=35 from UTSW and n=25 from Rockefeller), across segmentation methods.

The online version of this article includes the following figure supplement(s) for figure 2:

**Figure supplement 1.** Syllable onset detection accuracy across methods.

**Figure supplement 2.** Syllable offset detection accuracy for Rockefeller song library birds.

**Figure supplement 3.** Additional AVN labeling examples.

**Figure supplement 4.** AVN labeling consistency.

**Figure supplement 5.** Syllable labeling metrics across bird conditions.

find 3 different syllables clusters: one for 'a', one for 'b', and one for the combined 'ab'. Because of how we align syllables across segmentations for validation, syllable 'b' will sometimes carry one cluster label and sometimes carry the missing segment label, whereas syllable 'a' can carry two different clustering labels: one for 'a' and one for the 'ab' segmentation. This scenario would result in a lower completeness score (a measure of the extent to which syllables with the same manual annotation also carry the same cluster label) without any impact on the homogeneity score (a measure of the extent to which syllables with the same cluster label carry the same manual annotation). Consistent with this, Tweetynet segmentation results in higher homogeneity scores for typical birds than WhisperSeg, but lower completeness scores (*Figure 2—figure supplement 5*).

In cases where it is somewhat ambiguous as to whether syllables should be segmented separately ('a' and 'b') vs. merged ('ab'), WhisperSeg is much more likely to merge them than is Tweetynet, resulting in a lower homogeneity score when the manual annotation has the syllables segmented separately. However, WhisperSeg will be more consistent in its choice to merge or split than Twee-tynet within the same bird, which accounts for WhisperSeg's higher completeness score (*Figure 2—figure supplement 5*). WhisperSeg's internal consistency in this regard also results in less artifactual inflation of downstream syntax and syllable duration variability metrics, making it the better overall choice for syllable segmentation in the context of AVN.

Altogether, we conclude that WhisperSeg followed by UMAP-HDBSCAN clustering produces the most accurate syllable labels. These will be referred to as AVN labels henceforth in this manuscript. AVN labels are produced without the need for any per-bird parameter tuning or model training. This approach not only saves experimenters time when analyzing many birds but also reduces the potential for experimenter bias during song annotation. AVN labeling generalizes well across multiple zebra finch colonies, suggesting that it can be easily adopted by new research groups without the need for extensive additional validation. Thus, we hope it can serve as a standard for song annotations when manual annotation is not required.

## Analyzing song syntax

The automatically generated AVN labels can be used to visualize and quantify a bird's song syntax. Typical zebra finches produce syllables in a very predictable order, where the syllable type that a bird will sing can be reliably predicted based on the immediately preceding syllable type (*Scharff and Nottebohm, 1991*; *Hyland Bruno and Tchernichovski, 2019*). Many studies have found that manipulations of the neural circuitry underlying song learning and production can disrupt a bird's syntax, leading to more variable sequences (*Scharff and Nottebohm, 1991*; *Garcia-Oscos et al., 2021*; *Xiao et al., 2021*; *Tanaka et al., 2016*). Methods used to quantify these syntax disruptions vary across papers and research groups, making it impossible to directly compare the severity of disruptions. We propose a comprehensive suite of features to describe a bird's song syntax, which can all be calculated using the AVN Python package and AVN graphical application.

First, we developed a new song syntax visualization, called a *syntax raster plot,* which lets researchers view many song bouts' syllable sequences simultaneously (*Figure 3a*). We can also visualize syntax using a transition matrix, which gives the probability of a syllable type being produced, given the preceding syllable type (*Figure 3b*). We quantify the stereotypy of a bird's syntax by calculating the entropy rate of the transition matrix and find a strong correlation ($r=0.89$, $p<0.05$) between entropy rates calculated using AVN labels and manual annotations, showing that our AVN labels are sufficiently reliable to describe a bird's syntax stereotypy (*Figure 3c*). We also find the same statistical relationship between groups as with manual annotations, namely that birds with FP1 KD and isolate birds have significantly higher entropy rates than typical birds (One-way ANOVA $F(2, 32)=15.05$, Tukey HSD p-adj FP1 vs. typical <0.005, p-adj isolate vs. typical <0.005; *Figure 3d*; *Figure 3—figure supplement 1a–d*). Multiple studies have also found that neural song-circuit manipulations can induce a 'stutter' in birds, that is increase the rate of syllable repetitions in their songs (*Sánchez-Valpuesta et al., 2019*; *Norton et al., 2019*; *Kubikova et al., 2014*), so we've introduced two additional metrics to specifically look at the rate of syllable repetitions in a bird's song; the mean number of times a syllable is produced in a row each time it is sung (repetition bout length), and the CV of the number of syllable repetitions (CV repetition bout length; *Figure 3—figure supplement 1e–f*).

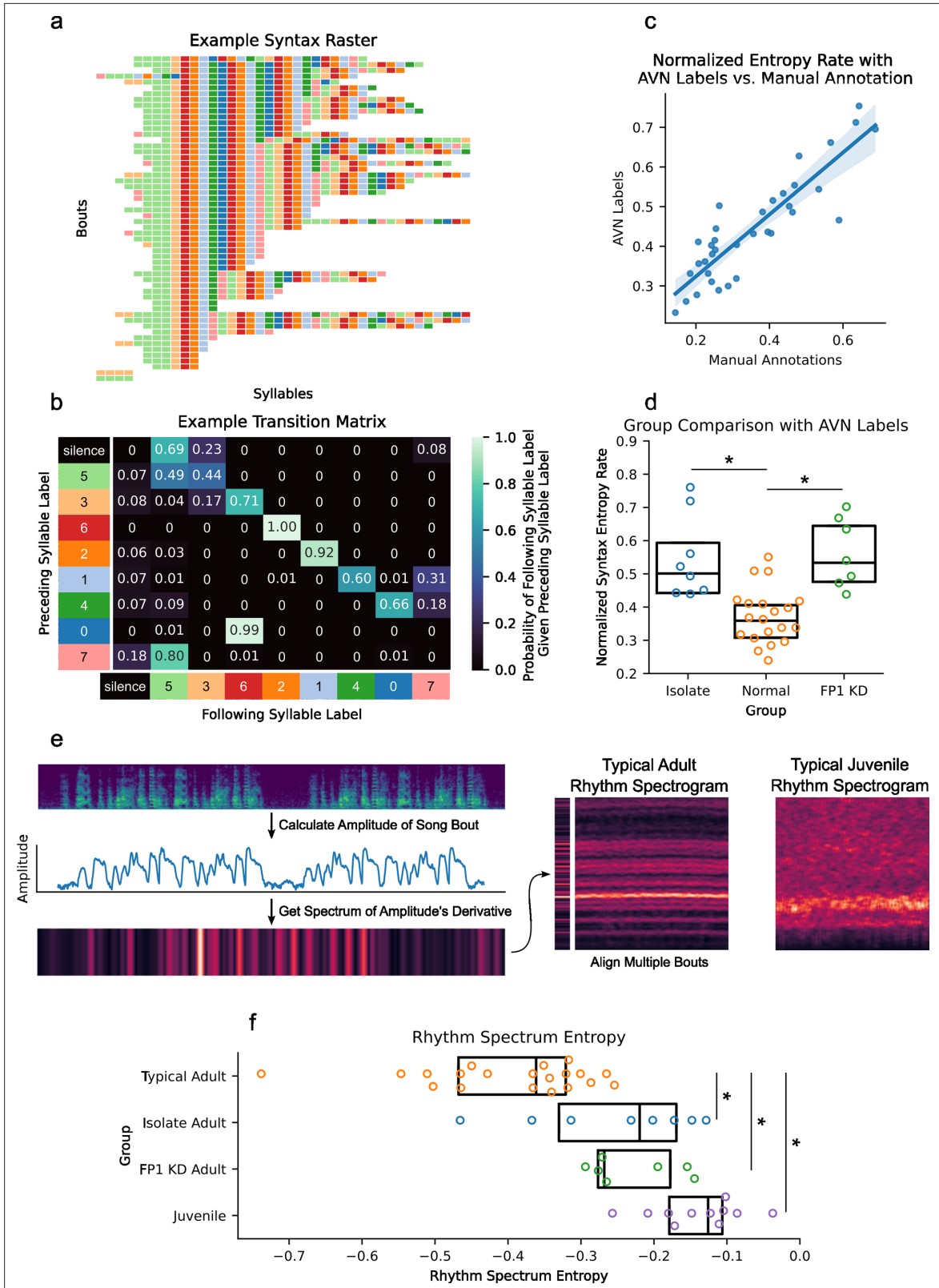

**Figure 3.** Song syntax and timing analysis with AVN. (**a**) Example syntax raster plot for a typical adult zebra finch made with AVN labels. Each row represents a song bout, and each colored block represents a syllable, colored according to its AVN label. (**b**) Example transition matrix from the bird featured in **a**. Each cell gives the probability of the bird producing the 'following syllable', given that they just produced a syllable with the 'preceding syllable' label. (**c**) Correlation between normalized entropy rate scores calculated for each bird using manual annotations or AVN labels (n=35 birds

*Figure 3 continued on next page*

*Figure 3 continued*

from UTSW, *r*=0.89, p<0.005). (**d**) Comparison of normalized entropy rates calculated with AVN labels across typical (n=20), isolate (n=8), and FP1 KD (n=7) adult zebra finches (One-Way ANOVA F(2, 32)=15.05, p<0.005, Tukey HSD * indicates p-adj <0.005). (**e**) Schematic representing the generation of rhythm spectrograms. The amplitude trace of each song file is calculated, then the spectrum of the first derivative of the amplitude trace is computed. The spectra of multiple song files are concatenated to form a rhythm spectrogram, with bout index on the x-axis and frequency along the y axis. The example rhythm spectrograms show the expected banding structure of a typical adult zebra finch, and the less structured rhythm of a typical juvenile zebra finch (50 dph). (**f**) Comparison of rhythm spectrum entropies across typical (n=20), isolate (n=8), FP1 KD (n=7) adult zebra finches (>90 dph), and juvenile zebra finches (n=11, 50-51 dph; One-Way ANOVA F(3, 43)=17.0, p<0.05, Tukey HSD * indicates p-adj <0.05).

The online version of this article includes the following figure supplement(s) for figure 3:

**Figure supplement 1.** Additional syntax examples and metrics.

**Figure supplement 2.** Syllable and gap duration entropy.

**Figure supplement 3.** Additional rhythm spectrogram examples.

## Analyzing song timing

Song timing can refer to the durations of individual syllables and gaps, or to the rhythmic patterns of a song bout. We have developed and validated multiple metrics to describe song timing at each of those scales, which can easily be calculated using the AVN Python package or graphical application. First, we look at the timing of individual syllables and gaps by plotting the distribution of their durations based on our WhisperSeg segmentations. Typical mature zebra finches have very stereotyped syllable durations across renditions of the same syllable type, which result in a distribution of syllable durations consisting of multiple narrow peaks, each corresponding to a different syllable type (***Figure 3—figure supplement 2***). Immature birds, on the other hand, have very variable syllable durations and tend to have a single broad peak and long positive tail in their syllable duration distributions (***Aronov et al., 2011***; ***Goldberg and Fee, 2011***). We observe these same patterns using our WhisperSeg segmentations when we apply them to our dataset of 35 mature birds, and an additional 11 juvenile birds aged 50–51 days post-hatch (dph). We quantify the maturity of a bird's syllable timing by calculating the entropy of their syllable duration distribution, which will approach 1 when density is evenly spread across syllable durations (as in juvenile birds), and approach 0 when density is concentrated in a narrow range of syllable durations, as was done in (***Goldberg and Fee, 2011***). Indeed, using our WhisperSeg segmentation, we find that juvenile birds have significantly higher syllable duration entropies than adult birds (F(3, 43)=17.43, p<0.005, Tukey HSD p-adj juvenile vs. typical adult <0.05; ***Figure 3—figure supplement 2***). The syllable duration entropy values that we obtain with WhisperSeg segmentation are also highly correlated with those scores that we obtain from manual segmentation (*r*=0.85, p<0.05; ***Figure 3—figure supplement 2***), further indicating that our automated segmentation is sufficiently accurate for downstream analyses.

In addition to syllable level timing, we have implemented multiple metrics describing a song's rhythm at the bout level based on the 'rhythm spectrogram' first proposed in ***Saar and Mitra, 2008***. A rhythm spectrum is constructed by taking the Fourier transform of the derivative of the amplitude trace of a song. If the song consists of multiple motifs with a consistent rhythm, the song's amplitude will have a repeating fluctuation pattern, which will be reflected in its spectrum. By calculating the 'rhythm spectrum' of multiple bouts, then concatenating them into a rhythm spectrogram, we can get a clear impression of a bird's overall rhythmicity, and the consistency of that rhythm across song bouts (***Figure 3e***). We quantify the strength of this rhythm by computing the Wiener Entropy of the mean rhythm spectrum, and we quantify the consistency of the rhythm across bouts by calculating the coefficient of variation of the peak frequency across each bout in the rhythm spectrogram (***Figure 3—figure supplement 3***). We find that juvenile birds have significantly higher rhythm spectrum entropies (***Figure 3f***, One-Way ANOVA F(3, 43)=17.0, Tukey HSD juvenile vs. typical adult p-adj <0.005) and higher peak frequency CVs than adult birds (***Figure 3—figure supplement 3***, One-Way ANOVA F(3, 43)=8.23, Tukey HSD juvenile vs. typical adult p-adj <0.05). Although the FP1 KD birds' syllable duration entropies are squarely in line with typical adults (***Figure 3—figure supplement 2e***, Tukey HSD FP1 KD vs typical adult p-adj=0.53), their rhythm spectrum entropies (***Figure 3f***) and gap duration entropies (***Figure 3—figure supplement 2f***) are significantly higher (Tukey HSD FP1 KD vs. typical adult p-adj <0.05). This is consistent with our earlier finding that the FP1 KD birds have more variable

syllable sequencing and highlights the complementary nature of these metrics. When considered together, they provide a comprehensive description of a bird's song production.

## Comparing song disruptions with AVN features

In addition to syntax and timing features, AVN can also calculate a suite of acoustic features, including goodness of pitch, mean frequency, frequency modulation, amplitude modulation, entropy, amplitude, and pitch. This feature set is well established for describing zebra finch song, thanks to the Sound Analysis Pro application (*Tchernichovski et al., 2000*). These features are calculated for each frame of a spectrogram, but to facilitate comparisons between birds, we take the mean value of each feature for every syllable rendition, then compute the overall mean value and the coefficient of variation across renditions of the same syllable type. We then select the syllable types with the minimum, median, and maximum values with respect to each feature to represent the overall acoustic properties of a bird's song. This results in a total of 48 acoustic features for each bird. When combined with our 3 syntax-related features and 4 timing-related features, we are left with a complete set of 55 features to describe all major aspects of a bird's song production. This feature set represents an extremely valuable resource for comparing experimental groups, for tracking song phenotypes over time, or for detecting birds with atypical song production.

To showcase the AVN feature set's potential for comparing birds across experiments and research groups, we calculated this feature set for 53 typical adult zebra finches, 16 isolate-reared zebra finches, and 7 FP1 KD zebra finches from the UTSW colony; 25 typical adult zebra finches from the Rockefeller Song Library (*Tchernichovski et al., 2021*) and 4-sham deafened birds and 5 early-deafened birds from Hokkaido University, originally recorded for (*Mori and Wada, 2015*). We fit a Linear Discriminant Analysis (LDA) model to a dataset containing only typical and isolate zebra finches. LDA finds the linear combination of features that best separates groups of points (in this case, typical vs. isolate birds' AVN features), allowing us to identify which features are most informative for discriminating between these groups. This allows us to calculate an 'isolate score' for every bird, based on the LD1 value (*Figure 4a*). We achieved a 95% classification accuracy between these two groups (*Figure 4a*), with the most important features being higher syntax entropy rates, higher syllable duration variance, and higher rhythm entropies for isolates compared to typical birds (*Figure 4—figure supplement 1*). We repeated this process for typical hearing and deaf birds and achieved a 99% classification accuracy (*Figure 4b*), with the most important features being higher mean frequency variance, lower absolute mean frequency, and higher syllable duration variance for deaf birds compared to hearing birds (*Figure 4—figure supplement 1*).

Finally, we fit an LDA model to a dataset containing typical, isolate, and deaf zebra finches and used this model to assess which of these groups the FP1 KD birds most closely resemble. Previous work suggests that FP1 KD in the song nucleus HVC impairs tutor song memory formation while leaving song motor learning intact (*Garcia-Oscos et al., 2021*). Thus, we would expect the FP1 KD birds' songs to most closely resemble those of isolates, who have no tutor song memory but do have access to auditory feedback of their own vocalizations, and to not resemble deaf birds who have neither tutor song memories nor access to auditory feedback. Indeed, we find that 5/7 FP1 KD birds are classified as isolates by the LDA classifier, with the other 2/7 being classified as typical birds (*Figure 4c and d*). This supports our hypothesis about the nature of the song disruption in FP1 KD birds and highlights the utility of a common feature set for comparing song phenotypes. We hope that the ease of calculation and completeness of this feature set will facilitate phenotypic comparisons across the field of songbird neuroscience, particularly as the field grows in its ability to conduct genetic manipulations of the neural circuits involved in different aspects of song learning and production.

## Tracking song development with AVN features

To further showcase the potential of these AVN features for zebra finch song analysis, we also used them to track song development. We calculated the AVN feature sets for 14 birds from UTSW and 5 birds from Duke University (*Brudner et al., 2023*) at multiple time points during song development, ranging from 46 dph to 102 dph. We fit a Generalized Additive Model (GAM) to predict a bird's age based on its AVN features and find that we achieve the most accurate age predictions with a model that considers a bird's syllable duration entropy, their syntax entropy, absolute syllable durations, and the variability of goodness of pitch, syllable duration, and Wiener entropy across renditions (*Figure 5*).

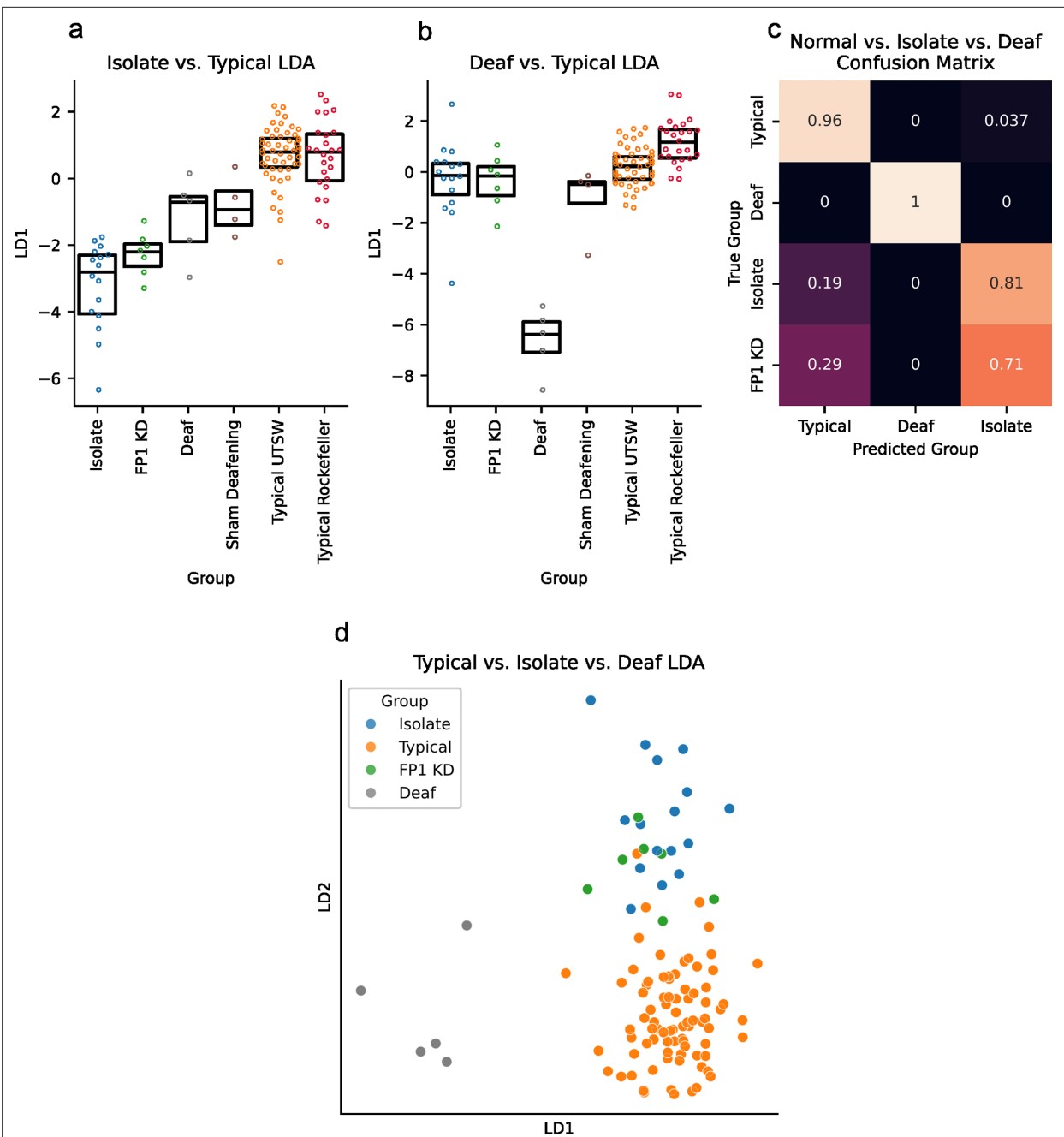

**Figure 4.** Song phenotypes classification with AVN features. (**a**) Linear discriminant values for multiple groups of birds generated from a model trained to discriminate between typical and isolate zebra finches (n=16 isolate birds, 7 FP1 KD birds, 5 deaf birds, 4 sham deafening birds, 53 typical zebra finches from the UTSW colony, and 25 typical zebra finches from Rockefeller). (**b**) Linear discriminant values for multiple groups of birds generated from a model trained to discriminate between typical and deaf zebra finches. Same birds as in **a**. (**c**) Confusion matrix indicating the LDA model's classification of typical, deaf, isolate, and FP1 KD birds from aa model trained to discriminate between typical, deaf, and isolate birds. Scores for typical, deaf, and isolate birds were obtained using leave-one-out cross-validation, and FP1 KD scores were obtained using a model fit to all typical, deaf, and isolate birds. (**d**) Plot of the linear discriminant coordinates of isolate (n=16), typical (n=78), and FP1 KD birds (n=7) for a model trained to discriminate between typical, deaf, and isolate birds. FP1 KD birds overlap most with isolate birds in this LDA space, indicating that their song production most closely resembles that of isolates.

The online version of this article includes the following figure supplement(s) for figure 4:

**Figure supplement 1.** Song phenotype classification model feature importances.

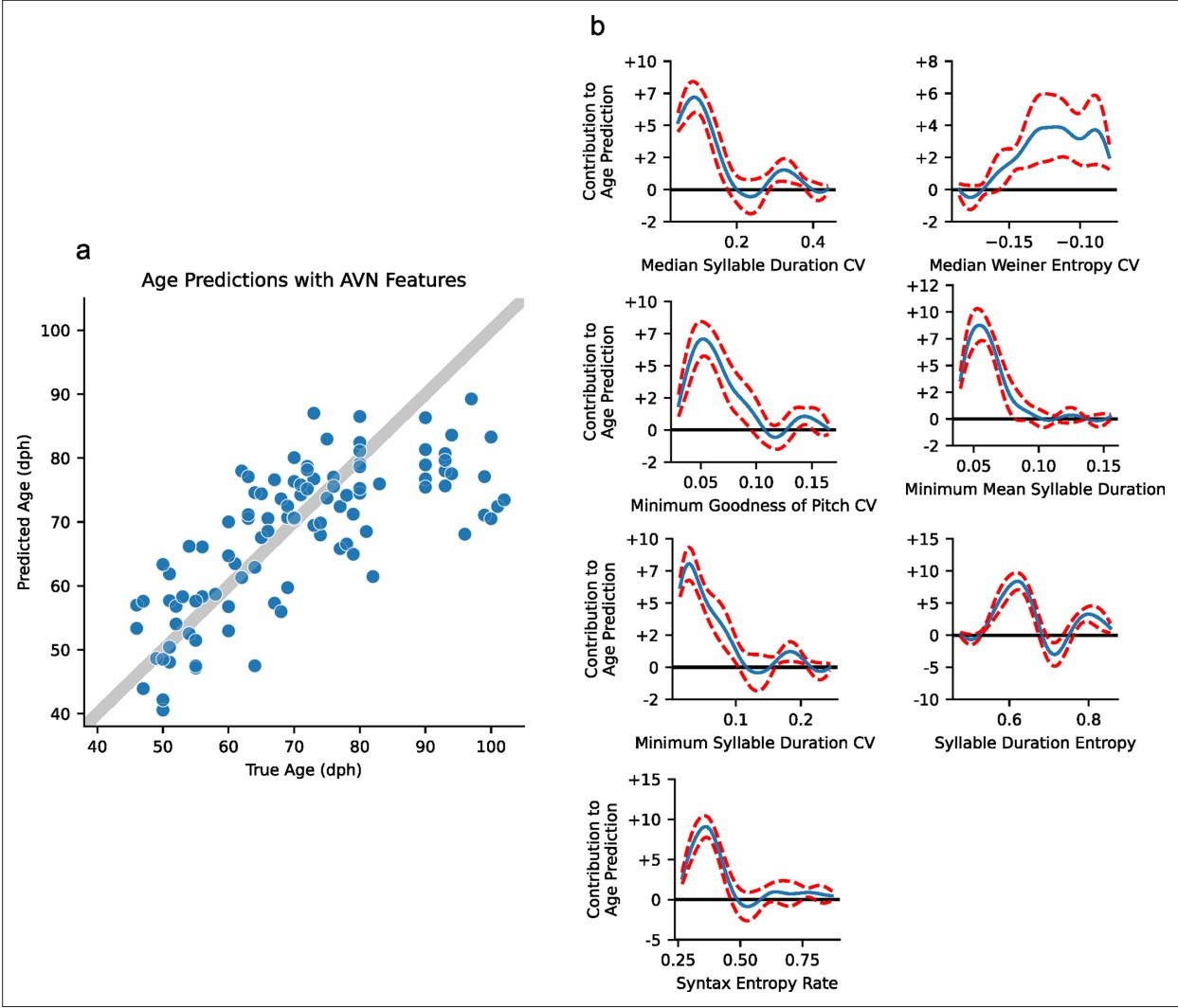

**Figure 5.** Age prediction with AVN features. (**a**) Generalized additive model's age predictions vs. true ages for 103 days of song recordings across 19 individual birds. Model predictions were generated using leave-one-bird-out cross-validation. The gray line indicates where points would lie if the model were perfectly accurate. (**b**) Partial dependence functions for each feature in the GAM model. The values of each feature along the x-axis map onto learned contributions to the age prediction along the y-axis. The GAM model's prediction is the sum of these age contributions based on each day of song's feature values, plus an intercept term.

When trained with data from all but one bird and tested on the remaining bird, this model can predict a bird's age within 7 dph for 50% of age points, and within 11 dph for 75% of age points. Its performance is best for younger birds, with prediction accuracy dropping considerably for birds over 80 dph, which is expected as song changes slow with age and eventually stabilize when birds reach around 90 dph. This could be a powerful tool to measure whether an experimental manipulation causes a bird's song maturity to systematically lead or lag their true biological age, and which song features are most responsible for this effect, particularly as this method doesn't require typical mature song from each subject bird to make its estimate, unlike *Goffinet et al., 2021*; *Brudner et al., 2023*, for instance.

## Measuring song imitation

So far, we've demonstrated how AVN's features can be used to describe and compare adult and juvenile song production across experiments and research groups. While these features are sufficient to predict a bird's song learning stage and to detect abnormalities in experimental groups, they don't directly reflect song learning success. Zebra finches learn song by imitating an adult tutor, and this song learning is typically assessed by comparing a pupil bird's song to its tutor's, with higher similarity

reflecting more successful learning (*Tchernichovski et al., 2000*, but see *Tchernichovski et al., 2021*). Many methods for zebra finch song similarity scoring currently exist; however, they all require either the manual identification of pupil and/or tutor motifs (*Tchernichovski et al., 2000*; *Mandelblat-Cerf and Fee, 2014*; *Lachlan, 2007*), which limits the number of renditions that can be considered and has the potential to introduce experimenter bias, or require retraining or re-calibration when applied to new tutor-pupil pairs (*Goffinet et al., 2021*; *Mets and Brainard, 2018*), which makes it impossible to directly compare learning outcomes across experiments.

To overcome these limitations, we have developed a novel similarity scoring system that doesn't require any manual motif identification, or any retraining or re-calibration for new tutor-pupil comparisons. Our approach involves a deep convolutional neural network that is trained with a dataset of over 16,000 manually annotated syllables from 21 adult zebra finches from the UTSW colony. These syllables are presented to the model in triplets, consisting of a randomly selected 'anchor' syllable, a 'positive' syllable which belongs to the same type as the anchor, and a 'negative' syllable, which belongs to a different syllable type. The model learns to map spectrograms of syllables to an eight-dimensional embedding space, such that the anchor syllable's embedding is closer to the positive's embedding than to the negative's. We use the trained network to compute the syllable embeddings for hundreds of syllables produced by a pupil bird and by its tutor and measure the similarity between their songs by calculating the Maximum Mean Discrepancy (MMD) between their syllable distributions (*Figure 6a*). Deep convolutional networks trained with triplet loss have been shown to be useful data compression steps before further downstream tasks are applied on their output, particularly in contexts where the data consists of many discrete classes (syllable types), and relatively few samples per class (renditions of each syllable type; *Thakur et al., 2019*; *Schroff et al., 2015*).

This process of learning a low-dimensional representation of syllables, then calculating repertoire similarity by comparing distributions of syllables in the low-dimensional space was inspired by *Mets and Brainard, 2018*, and is conceptually similar to the method proposed in *Goffinet et al., 2021*, who use a variational autoencoder (VAE) to compute their syllable embeddings, followed by MMD to score syllable distribution similarity. As such, we've compared the similarity scoring results we obtain with our triplet loss model and MMD to *Goffinet et al., 2021*'s VAE and MMD, as well as each of these methods combined with another empirical distribution comparison metric: Earth Mover's Distance (EMD).

We first validated these approaches with a dataset of 30 typical tutor-pupil pairs from the UTSW colony segmented using WhisperSeg, none of whom share a song tutor with any of the birds used to train the triplet loss model or VAE. The triplet loss model and VAE consistently yielded higher MMD and EMD dissimilarity scores between a pupil and unrelated bird, compared to a pupil vs. another bird with the same tutor (a 'sibling') and a pupil vs. its tutor, as expected (*Figure 6b*; *Figure 6—figure supplement 1a–d*). To more quantitatively compare the sensitivity of each of these approaches in distinguishing different types of relationships between typical zebra finches, we computed a contrast index and tutor contrast index score for each pupil bird. The contrast index represents the difference in dissimilarity score between two subsets of syllables from the same pupil bird, and between the syllables of the given pupil bird and three randomly selected unrelated birds. We find that the VAE with MMD yields the highest contrast index score (contrast index = 0.967 + SEM 0.009), and triplet loss with MMD as a close second (contrast index = 0.938 + SEM 0.015; *Figure 6—figure supplement 1e*). Notably, all the methods result in much higher average contrast index scores than Sound Analysis Pro (*Tchernichovski et al., 2000*; contrast index = 0.156) and (*Mandelblat-Cerf and Fee, 2014*; contrast index = 0.41).

Traditional contrast index doesn't paint a complete picture of the reliability of a method for similarity scoring, however. Typically, similarity scores are computed between a pupil bird and its tutor to measure the pupil's song learning success. In cases of successful song learning, there will be a high degree of similarity between a pupil's song and their tutor's, but these will still be much more different than two subsets of a pupil's song compared to each other, as is used to calculate contrast index. A useful similarity scoring method must therefore have sufficient sensitivity among higher dissimilarity comparisons to distinguish between imitated song (a pupil vs. tutor comparison) and non-imitated song (a pupil vs. an unrelated bird). This is precisely what is measured by the tutor contrast index, which was highest for the triplet loss model and MMD (tutor contrast index = 0.316 + SEM 0.035;

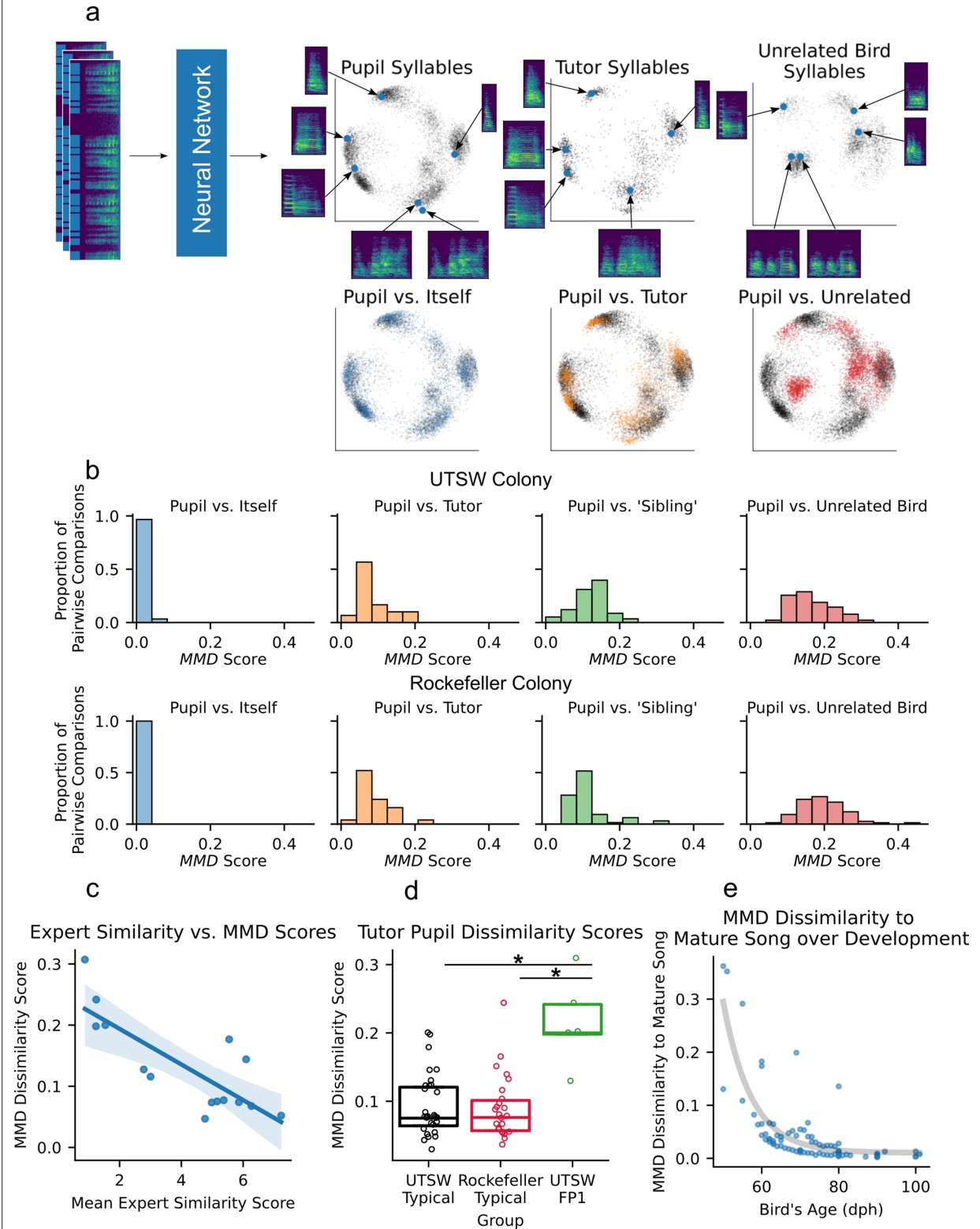

**Figure 6.** Illustration and validation of AVN's song similarity scoring method. (**a**) Schematic of the similarity scoring method. A deep convolutional neural network is used to embed syllables in an 8-dimensional space, where each syllable is a single point, and similar syllables are embedded close together. The first two principal components of the eight-dimensional space are used for visualization purposes only here. The syllable embedding distributions for two random subsets of syllables produced by the same pupil on the same day have a high degree of overlap. The empirical distributions of all syllables from a pupil and his song tutor are less similar than a pupil compared to himself, but still much more similar than a pupil and a random

*Figure 6 continued on next page*

*Figure 6 continued*

unrelated bird. (**b**) Maximum Mean Discrepancy (MMD) dissimilarity score distribution for comparisons between a pupil and itself (n=30 comparisons for UTSW, n=25 for Rockefeller), a pupil and its tutor (n=30 comparisons for UTSW, n=25 for Rockefeller), two pupils who share the same tutor (aka pupil vs. 'Sibling' comparisons, n=58 comparisons for UTSW, n=64 for Rockefeller), and between two pupils who don't share song tutor (aka pupil vs. unrelated bird, n=90 comparisons for UTSW, n=75 for Rockefeller). Calculated with a dataset of 30 typical tutor-pupil pairs from UTSW and 25 from Rockefeller. (**c**) Correlation between MMD dissimilarity scores and human expert judgements of song similarity for 14 tutor-pupil comparisons from the UTSW colony ($r$=–0.80, $p<0.005$). (**d**) Tutor-pupil MMD dissimilarity scores for typical pupils from the UTSW colony (n=30), typical pupils from the Rockefeller Song Library (n=25), and FP1 KD pupils from the UTSW colony (n=7) (One-Way ANOVA $F(2, 57)=9.57$, $p<0.005$). * Indicates Tukey HSD post hoc p-adj<0.05). (**e**) MMD Dissimilarity score between birds at various age points across development, compared to their mature song recorded when the bird is over 90 dph. Each point represents one comparison (n=91 comparisons across 11 birds). Gray line is an exponential function fit to the data to emphasize the slowing of song maturation as birds approach maturity.

The online version of this article includes the following figure supplement(s) for figure 6:

**Figure supplement 1.** Comparison of embedding and distance metric methods for similarity scoring.

**Figure supplement 2.** Additional similarity scoring validation.

**Figure supplement 3.** Similarity scoring deep neural network architecture.

*Figure 6—figure supplement 1f*). Therefore, our triplet loss model combined with MMD is the most sensitive method for evaluating song learning.

The triplet loss model with MMD dissimilarity scores shows the same pattern and absolute range of scores for pupil vs. self, pupil vs. tutor, pupil vs. 'sibling', and pupil vs. unrelated bird comparisons for the UTSW dataset and for a dataset of 25 tutor-pupil pairs from the Rockefeller Song Library, despite these birds being recorded under different conditions from any of the birds used in model training (*Figure 6b*; *Figure 6—figure supplement 2*). This shows that the trained model generalizes well to birds from other research groups without the need for any additional fine-tuning and thus can serve as a standard approach for the field. This approach also yields similarly high contrast indices and tutor contrast indices for both the UTSW and Rockefeller datasets (*Figure 6—figure supplement 2*, UTSW mean contrast index = 0.938 + SEM 0.015, Rockefeller mean contrast index = 0.920 + SEM 0.020, t-test p=0.47; UTSW mean tutor contrast index = 0.316 + SEM 0.035, Rockefeller mean tutor contrast index = 0.407 + SEM 0.031, t-test p=0.06). MMD scores produced by this model also agree better with expert human judgements of song similarity than do %similarity scores calculated with Sound Analysis Pro (*Figure 6c*; MMD vs. human expert absolute $r$=0.80, *Figure 6—figure supplement 2b*, SAP %similarity vs. human expert absolute $r$=0.33).

Using this method, we find that FP1 KD pupils have significantly higher MMD dissimilarity to tutor scores when compared to typical birds from either UTSW or Rockefeller (*Figure 6d*; One-Way ANOVA $F(2, 57)=9.6$, Tukey HSD FP1 KD vs typical UTSW p-adj<0.005, FP1 KD vs typical Rockefeller p-adj<0.005), showing that this method can be used to assess song learning outcomes in experimentally manipulated birds. We also used the model to look at how a bird's song changes over development, by comparing song at multiple age points to a bird's mature song. As expected, we find that birds gradually become more similar to their mature song over the course of development, and that the rate of this change slows as birds approach maturity (*Figure 6e*). Altogether, these tests showcase that this method (triplet loss model embedding with MMD) is more reliable at assessing tutor-pupil song similarity than existing methods, while also not requiring any manual motif identification or dataset-specific fine-tuning. As a result, as with the AVN acoustic, timing, and syntax features, its scores are directly comparable across research groups, facilitating the quantitative comparison of song learning outcomes across studies.

## Discussion

Here, we have presented the AVN song analysis pipeline, which performs highly accurate syllable segmentation and syllable labeling. We have shown that this approach yields consistently high performance across multiple zebra finch colonies, suggesting that it can standardize and simplify large-scale behavioral annotation across research groups, without the need for additional training or fine-tuning. The AVN labels are used to calculate syntax features which agree well with manual annotations, and which are sufficient to discriminate between typical birds and birds with known genetic disruptions. The AVN segmentations and raw song files are used to calculate timing features, which again are

consistent across colonies and which reflect a bird's stage in song development. Standard acoustic features are also calculated for each AVN syllable type, which can be used to describe the overall acoustic properties of a bird's song.

To showcase the utility of these song features, we presented how they can be used to compare multiple different song phenotypes, to test our hypothesis that the songs of FP1 KD birds would more closely resemble isolate birds' compared to typical or deaf birds' songs (*Garcia-Oscos et al., 2021*). We also showed how these features can be used to create an interpretable model to predict a bird's age within 7 days while their song is rapidly evolving from immature subsong to stable adult song. As more research groups use the AVN feature set to describe their birds' song phenotypes, these analyses will only become more sensitive and powerful. Ultimately, we hope that these song features can be used to establish a comprehensive map of song phenotypes, which more closely link abnormal song phenotypes with the neural circuit dysfunctions underlying them.

Finally, we developed a novel similarity scoring system that outperforms existing methods in its sensitivity and fidelity to expert human judgements of song similarity, all without requiring any manual song annotation. Again, we expect this to be an invaluable tool for describing the nature and severity of song learning phenotypes in experimentally manipulated birds, where existing similarity scoring methods perform particularly poorly.

AVN is available to researchers as an open-source Python package and as a graphical application. The Python package allows researchers with some coding experience to take full advantage of the flexibility of these tools and integrate this pipeline into their data collection and processing workflows, while the application allows other researchers to easily annotate their songs and calculate AVN features with minimal coding, in a highly reproducible fashion.

Altogether, we see this pipeline as an example of the integration of deep learning tools and expertly curated features to automated behavior analysis without compromising the interpretability or generalizability of results. This feature set and annotation approach was designed with zebra finches in mind, but should be easily adaptable to other species with discrete syllables that can be clustered according to their acoustic features, such as Bengalese finches and Canaries, for example *Sainburg et al., 2020*. These species have more complex syllable sequencing than zebra finches and would therefore also benefit from additional syntax and timing features specific to their species. Additionally, while we've strived for a comprehensive set of features, it is possible that our 55-feature set will fail to reflect certain interesting song phenotypes that haven't yet been observed. We hope that the open-source nature and extensive documentation of the AVN pipeline will allow and encourage researchers to contribute additional song features to the pipeline as they encounter such cases where the current feature set may be insufficient.

## Methods

The AVN documentation, AVN-GUI, and code necessary to produce all figures in this manuscript can be found through the following links:

AVN documentation: https://avn.readthedocs.io/en/latest/index.html
AVN-GUI: https://avn.readthedocs.io/en/latest/AVN_GUI.html
Code for figures: https://github.com/theresekoch/AVN_paper, copy archived at *Koch, 2025a*
Model training: https://github.com/theresekoch/Triplet_loss_model_training (*Koch, 2025b*)

### Animals

Experiments described in this study were conducted using male zebra finches. All procedures were performed in accordance with protocols approved by Animal Care and Use Committee at UT Southwestern Medical Center, protocol number 2016–101562 G.

### Data acquisition

A complete list of birds and the analyses in which they were included can be found in *Supplementary file 1*.

### UTSW dataset

Many birds included in this study were previously recorded and analyzed in *Garcia-Oscos et al., 2021*. This includes seven birds which were injected with a pscAAV-GFP-shFoxP1 virus before exposure to a

song tutor, leading to disrupted songs (referred to as FP1 KD birds in this manuscript and FP1-KD SE in *Garcia-Oscos et al., 2021*). Eight birds were included in this group in the previous paper. One was omitted from this manuscript because it exhibited completely typical song, likely due to weak viral expression. A further 10 birds which were injected with a control virus before exposure to a song tutor (Ctrl SE in *Garcia-Oscos et al., 2021*), and 10 which were injected with the pscAAV-GFP-shFoxP1 after tutor song exposure (FP1-KD BI in *Garcia-Oscos et al., 2021*) are included in the current study. Both of these groups exhibit species-typical song production and are included in the 'typical' group in this study. Finally, 8 additional birds which were raised in isolation from an adult song model until they were at least 90 days post-hatch ('Full isolates' in the FP1 paper and 'Isolates' in the current study) were included in this study, for a total of 35 birds. See *Garcia-Oscos et al., 2021* for more information on viral injections and rearing conditions.

An additional 8 adult isolate birds, 37 typically reared adult birds, and 10 juvenile birds were recorded for this study. For isolated birds, fathers were removed from breeding cages before the young reached 12 dph. These young remained housed with their mother and siblings in a room containing only other isolate breeding cages, until they were weaned between 40 and 60 dph, at which point they were housed individually, with auditory but no visual access to other isolate reared males. Typically, reared and juvenile birds were raised in our main colony room which contained about 55 breeding pairs. They had unlimited access to their father's song in their home cages, until they were weaned between 40 and 60 dph, at which point they were housed in group cages with other males.

Recordings were obtained by individually housing birds in sound-attenuating chambers. They were continuously recorded using Sound Analysis Pro 2011 (*Tchernichovski et al., 2000*). All birds were placed on a 14 hr:10 hr day:night cycle and provided ad libitum access to food, water, and grit. All procedures were performed in accordance with protocols approved by the Animal Care and Use Committee at UT Southwestern Medical Center.

## Additional song data

In addition to the birds recorded at UTSW, this study includes recordings of 25 pupils and 6 tutors from the Rockefeller University Field Research Center Song Library (*Tchernichovski et al., 2021*), 5 juvenile birds from Duke University (*Brudner et al., 2023*), and 5 early deafened and 4 sham deafened birds from Hokkaido University (*Mori and Wada, 2015*). See citations for more information on rearing and recording conditions.

## Manual song annotation

A random subset of 30 song files from a single day of recording was annotated for each of 35 adult birds from UTSW, and 15 song files were annotated for each of 25 adult birds from the Rockefeller Song Library. Manual annotation was performed using the *evsonganaly* application in MATLAB (*Tumer and Brainard, 2007*), and involved (1) amplitude threshold syllable segmentation with a threshold selected for each song file based on visual inspection of the amplitude trace and spectrogram, (2) manual correction of erroneous syllable onsets or offsets, and (3) assignment of syllable labels to each syllable based on visual inspection of the spectrogram. All annotations were prepared by one of two expert annotators. These annotators consulted with each other in cases of ambiguous syllable segmentation or labeling and used the same existing set of labeled songs as a reference. Both annotators agreed to adopt the more conservative approach of splitting syllables into smaller segments in cases of ambiguous segmentation and assigning different syllable labels in cases of ambiguous labeling.

For all applications except training TweetyNet (*Cohen et al., 2022*), segments that reflect cage noise were dropped from the annotations based on visual inspection of the spectrograms. A second set of annotations was made which retained noise segments, labeling them as such, with all syllables and calls labeled as simply 'vocalizations' for the purpose of training TweetyNet.

## Segmentation

### Amplitude segmentation

Amplitude segmentation was performed using the 'RMSEDerivative' class in the AVN Python package's segmentation module. Each song file is bandpass filtered between 200 Hz and 9000 Hz, then the root mean square energy (RMSE) of each audio frame is computed with a hop length of 512 samples

and a frame length of 2048 samples. The RMSE values of each song file are normalized, then the RMSE's first derivative is compared against user-specified thresholds. A syllable onset is identified as a positive crossing of the 'onset' threshold. Syllable offsets tend to be marked by more gradual changes in RMSE compared to syllable onsets, making it difficult to identify them consistently. To mitigate this, we perform onset to onset segmentation with this method, meaning each segment included a song syllable and the silent gap that immediately followed it. If a syllable onset is not followed by another onset within 300 ms (as in the end of a song bout), the offset is set as the first negative crossing of an 'offset' threshold after the syllable onset.

In keeping with the TweetyNet and WhisperSeg segmentation methods which don't require per-bird parameter adjustments, the same 'onset' and 'offset' thresholds were used for all birds in the dataset. These thresholds were selected using AVN's segmentation.Utils.threshold_optimization_many_birds() function, which compares the F1 scores relative to manual segmentation obtained with multiple different threshold values to identify the threshold value that results in the lowest mean F1 score across all 35 UTSW birds used for amplitude segmentation validation. The same thresholds selected based on the UTSW birds were used to segment the 25 Rockefeller Song Library birds, as a test of the generalization of this method without the need for manual segmentations.

## TweetyNet

The *vak* Python package was used to prepare datasets for, train, and generate segmentation predictions with the TweetyNet model (*Cohen et al., 2022*). TweetyNet is a deep neural network consisting of a block of convolutional layers followed by a bidirectional long short-term memory (LSTM) layer. The model takes a 1 s spectrogram of song as input, and labels each frame within that spectrogram. TweetyNet was designed for simultaneous syllable labeling and segmentation, in which case it would label each frame of the spectrogram with a syllable label or as silence. However, to make this model generalize to new birds without any additional training data, we instead trained TweetyNet to label each frame as a vocalization, silence, or noise (common sources of noise include the bird hopping around its cage and flapping its wings), rather than a more specific syllable type. When trained with such data from many birds, it can learn to distinguish vocalizations from noise and silence in a sufficiently general manner that the model can be applied to previously unseen individuals.

Manually annotated song files with label classes 'noise' and 'vocalization' were used to train TweetyNet in a leave-one-out cross-validation scheme, meaning the model was trained with data from all but one bird and tested on the withheld bird for each of the 35 birds in the UTSW dataset. A model trained with data from all 35 UTSW birds was used to segment the 25 validation birds from the Rockefeller Song Library dataset, to test the model's ability to generalize to new colonies. Full model training and prediction procedures can be found in this paper's accompanying GitHub repository. For more information on the TweetyNet model itself, see *Cohen et al., 2022*.

## WhisperSeg

WhisperSeg is an instance of the Whisper Transformer model which was pre-trained for automatic human speech recognition and fine-tuned for animal voice activity detection with a multi-species animal vocalization dataset (*Gu et al., 2023*). It takes a spectrogram representation of up to 2.5 s of song as input and outputs the indices of vocalization onsets and offsets in the spectrogram. These indices are then converted to timestamps, and a consistent labeling scheme for an entire song file is achieved through a 'majority-vote' post-processing step across overlapping 2.5 s song segments. Syllable segmentation was performed using the *whisperseg-large-ms-ct2* model, with hyperparameters optimized for zebra finch song segmentation, based on *Gu et al., 2023*. This was done using scripts shared in this paper's accompanying GitHub repository, which must be installed separately from AVN according to the instructions provided in the repository. For more information on the WhisperSeg model and its training, see *Gu et al., 2023*.

## Validation

Segmentation methods were compared on the basis of their precision, recall, and F1 scores relative to manual annotations.

$$\text{Precision} = \frac{\text{True Positives}}{\text{True Positives} + \text{False Positives}}$$

$$\text{Recall} = \frac{\text{True Positives}}{\text{True Positives} + \text{False Negatives}}$$

$$\text{F1} = \frac{\text{True Positives}}{\text{True Positives} + \frac{1}{2}\left(\text{False Positives} + \text{False Negatives}\right)}$$

where a true positive is a syllable onset in the automatic segmentation that is within 10 ms of a syllable onset in the manual annotation, a false positive is a syllable onset in the automatic segmentation that doesn't match with a syllable onset in the manual annotation within 10 ms, and a false negative is a syllable onset that is present in the manual annotation which doesn't match with an automatic segmentation onset within 10 ms. When determining onset alignments, we ensure that each syllable onset in the manual annotation can only 'match' with a single syllable onset in the automatic segmentation, and vis-versa. This was done using AVN's 'segmentation.Metrics.calc_F1()` function. Across all three metrics, scores closer to 1 indicate better agreement between automatic and manual segmentations. These same features were calculated for syllable offsets as well, but with an allowance of 20 ms rather than 10 ms, to account for the greater variability in exact offset segmentation across all methods tested.

To further examine the temporal precision of each method relative to manual annotation, we also calculated the time difference in milliseconds between matched syllable onsets and offsets between automatic segmentations and manual annotations. This was done using AVN's 'segmentation.Metrics. get_time_delta_df()' function.

## Labeling
### UMAP dimensionality reduction
UMAP dimensionality reduction (*McInnes et al., 2018*) is performed on spectrograms of syllables, prior to HDBSCAN clustering for label assignment (*McInnes and Healy, 2017*). First, spectrograms of each segmented syllable are produced. The audio is first bandpass filtered between 500 Hz and 15 KHz, then amplitude normalized independently for each syllable rendition. The short-term Fourier transform of the normalized audio for each syllable is computed with a window length of 512 samples and a hop length of 128 samples. The resulting amplitude spectrogram is converted to decibels using the librosa 'amplitude_to_db()' function (*McFee et al., 2023*). The db-scaled spectrogram is then padded to match the dimensions of the longest syllable in a given bird's dataset, or clipped to 870 ms if it exceeds that duration. This limit is very generous, and only ever applies to segmentation errors, but it is necessary to avoid memory issues during UMAP computation. Normalized spectrograms are then flattened from an array to a single long vector, and the vectors corresponding to each spectrogram are concatenated into an array separately for each bird. This spectrogram array is used to calculate the UMAP embeddings of each syllable using the UMAP Python package's 'UMAP()' function. This approach is based on *Sainburg et al., 2020*.

At a high level, UMAP dimensionality reduction involves constructing a graphical representation of the syllable set, where each syllable spectrogram can be thought of as a point in high dimensional space which is connected to other syllables near it by edges. These edges are weighted based on the distance between points and the local density of those data points. The high-dimensional graph is then projected into lower dimensions in a way that best preserves its overall structure.

UMAP dimensionality reduction can be a useful initial step when attempting to cluster high-dimensional data points because many clustering algorithms, especially density-based clustering algorithms such as HDBSCAN, can suffer from the 'curse of dimensionality'. When clustering spectrograms directly, each pixel in the spectrogram is a dimension, meaning each spectrogram exists as a point in a space with thousands of dimensions. In such a high-dimensional space, points will be very sparsely distributed, even if the spectrograms appear largely very similar. As a result, it is very difficult to detect regions of higher point density to serve as the basis of clusters. Reducing the dimensionality of the dataset forces points closer together, such that regions of high density separated by lower density can be more easily detected. UMAP is particularly adept at emphasizing local clusters in high-dimensional data because of how its initial embedding graph is constructed.

UMAP parameters were selected based on suggestions in the UMAP-learn documentation for clustering using UMAP embeddings and based on visual inspection of plots and labeling outcomes compared to manual annotations for birds from the UTSW dataset.

## HDBSCAN clustering

The 'Hierarchical Density-based Spatial Clustering of Applications with Noise' (HDBSCAN) clustering algorithm (*McInnes and Healy, 2017*) was applied to the UMAP embeddings of syllable spectrograms for each bird independently, in order to assign syllable labels. This method was selected based on the results in *Sainburg et al., 2020* for clustering Bengalese finch syllables, a species closely related to zebra finches.

Essentially, HDBSCAN works by calculating the 'mutual reachability' distance between points in the UMAP space, based on the distance between them and their local densities. These mutual reachability distances serve as edges connecting nodes (points representing individual song syllables) in a graph, which are then pruned to obtain a minimum spanning tree (a graph using the minimum number of total edges to connect all points). The minimum spanning tree is then converted to a hierarchy by sorting the edges on the basis of their mutual reachability scores. Clusters of points are identified by defining a minimum cluster size and selecting the clusters that persist over the longest span of the hierarchy. As with the UMAP parameters, the same HDBSCAN hyperparameter set was used for all birds. The hyperparameter values were selected based on v-measure scores and visual inspection of confusion matrices for WhisperSeg segments compared to manual annotations for birds from the UTSW colony.

## Validation

Syllable labeling was assessed by comparing automatically assigned syllable labels to manual annotations. Automatically labeled segments first had to be aligned to the manual annotations to identify pairs of labels in the automatic clustering and manual annotation that referred to the same vocalization. This was done using the same method described in the *Segmentation Validation* section, in which syllable onsets are uniquely matched to their closest counterpart across segmentation methods, this time up to a maximum distance of 100 ms. False positive syllable detections (i.e. syllable present in the automatic segmentation without a manual annotation counterpart) are assigned to their own manual annotation category ('x'), and false negative syllable detections (i.e. syllables present in the manual annotation without an automatic segmentation counterpart) are assigned to their own cluster ('1000') for the purposes of visualization and quantification.

Once syllables have been aligned between automatic segmentation and manual annotations, the HDBSCAN cluster labels are compared to manual labels for each bird to construct a confusion matrix, which gives the number of syllables in each HDBSCAN cluster that carry each of the possible manual labels. The confusion matrix values can then be used to compute homogeneity, completeness, and v-measure scores, to evaluate the correspondence between HDBSCAN labels and manual annotations for each bird. Homogeneity measures the extent to which syllables with the same AVN label also carry the same manual annotation label, completeness measures the extent to which syllables with the same manual annotation label also carry the same AVN label, and the V-measure is the harmonic mean of these two scores.

$$\text{Homogeneity } = \frac{H\left(\text{manual labels}\,|\,\text{clusters}\right)}{H\left(\text{manual labels}\right)}$$

$$\text{Completeness} = \frac{H\left(\text{clusters}\,|\,\text{manual labels}\right)}{H\left(\text{clusters}\right)}$$

$$\text{V-measure} = \frac{2 \times \text{Homogeneity} \times \text{Completeness}}{\text{Homogeneity} + \text{Completeness}}$$

where $H\left(\text{manual labels}\,|\,\text{clusters}\right)$ is the conditional entropy of the manual labels given the cluster labels, $H\left(\text{manual labels}\right)$ is the entropy of the manual labels, and vis-versa. In all cases, a higher score indicates better correspondence between clusters and manual labels, with a maximum possible score of 1 and minimum score of 0.

As UMAP embeddings are inherently stochastic, the subsequent HDBSCAN labels can vary over repeated runs with the exact same syllable dataset, depending on the random initialization of the UMAP embedding. To examine how consistent the labels are across different random UMAP initializations, we calculated the v-measure scores for 35 birds from the UTSW colony across 30 different random initializations and calculated the standard deviation of the v-measure scores for each bird, as a measure of the outcome variability due to UMAP's stochasticity.

## Syntax features

### Syntax raster plot

Beginning with a table of all AVN labels, syllables that are preceded and followed by a period of silence longer than 200 ms are removed, as they likely reflect calls produced outside of song. Song bouts are then identified as sequences of at least two syllables that are separated by silent gaps no longer than 200 ms. These bouts are aligned based on a user-specified alignment syllable, such that the first instance of the alignment syllable is in the same position across all bouts. This alignment is important as bouts typically begin with a variable number of introductory notes, which will obscure patterns in syllable sequence across bouts when they are not aligned to the first non-introductory note syllable. After alignment, bouts are ordered such that bouts with similar sequences after the alignment syllable are together in the final plot, which also helps emphasize patterns across bouts. This is done using AVN's syntax. Syntax_Data.make_syntax_raster() function. See avn's documentation for additional information and examples.

### Syllable transition matrix

As with syntax raster plots, syllables that are preceded and followed by more than 200 ms of silence are dropped from the AVN labels as they likely reflect calls produced outside of song. Silent gaps longer than 200 ms and file bounds are then added as states to the AVN label sequence. All syllable transitions between AVN labels are then counted, including transitions to and from periods of over 200 ms of silence; meaningful transitions as they reflect the beginnings or ends of song bouts. Transitions to and from file bounds are ignored, as these are artifacts of the recording and don't reflect meaningful behavioral states. The transition counts are then divided by the total number of renditions of the first syllable type in the transition to get the conditional probability of the second syllable, given the first syllable. This is done using AVN's syntax.Syntax_Data.make_transition_matrix() function.

### Syntax entropy rate

Syntax stereotypy is quantified using the entropy rate of the syllable transition matrix.

$$\text{entropy rate} = -\sum_{i,k} \pi_i p_{i,k} \log_2\left(p_{i,k}\right)$$

where $p_{i,k}$ is the probability of transitioning from initial syllable type $i$ to following syllable type $k$, and $\pi_i$ the probability of syllable type $i$ occurring, regardless of what syllable precedes or follows it. An entropy rate approaching 0 indicates that all transitions are highly predictable. The maximum possible entropy rate score is $log_2\left(N\right)$ where $N$ is the number of syllable types in the bird's repertoire plus one to account for silence as a possible state. To directly compare scores between birds without being biased by the number of syllable types in their song (a feature which depends strongly on the number of syllable types present in their tutor's song), we divide the entropy rate score by $log_2\left(N\right)$ such that it is now bounded between 0 and 1. This is done using AVN's syntax.Syntax_Data.get_entropy_rate() function.

### Repetition bouts

A repetition bout refers to every instance in which a syllable is produced, either a single time or multiple times in a row. For example, in the syllable sequence *abcaaabc*, syllable *a* has two repetition bouts, one of length one, meaning the syllable was produced without being repeated, and one of length 3, meaning the syllable was produced three times in a row. The number and length of repetition bouts are calculated for each syllable type in a bird's repertoire. The mean repetition bout length and coefficient of variation (CV) of repetition bout length are then calculated for each syllable type.

To facilitate comparisons across birds, which have different numbers of syllable types, the mean repetition bout length and CV of repetition bout length for the syllable type with the highest mean repetition bout length are selected to represent the bird's overall tendency to repeat syllables, excluding syllable types that reflect putative calls or introductory notes. Typical zebra finches often repeat calls or introductory notes, but rarely repeat song syllables, so looking at the repetition of song syllables is more informative when detecting or comparing birds with abnormal syntax. That said, in certain experiments, repetition bout features of calls or introductory notes may be of greater interest, in which case they can also be specifically identified using AVN.

An AVN syllable type is considered a putative introductory note if it (1) is no less than 5% less likely to be transitioned to from silence than the syllable type most commonly transitioned to from silence, meaning it tends to occur at the start of a vocalization bout and (2) it has a single dominant transition to a syllable type other than itself which is not silence, meaning that after a number of repetitions, it is eventually followed by a predictable next syllable type, which should reflect the start of a motif. These criteria were determined based on inspection of the syntax properties of introductory notes in manual song annotations. An AVN syllable type is considered a putative call if it is (1) not a putative introductory note and (2) produced in a bout of one or two syllables preceded and followed by at least 200 ms of silence in more than ⅓ of all utterances. This criterion was again determined based on visual inspection of manual song annotations.

## Song timing features

### Syllable and gap duration entropy

A syllable duration distribution is constructed based on the segment durations output by WhisperSeg for each bird. A histogram of the $\log_{10}$ of syllable durations is calculated, with 50 evenly spaced bins ranging from –2.5 to 0. As in *Goldberg and Fee, 2011*, the log of syllable durations is used because the syllable duration distributions of juvenile birds are roughly exponential and therefore linear in log space. Histograms are normalized to produce a probability density function across syllable durations. The entropy of this distribution is then calculated as

$$\text{Entropy} = \frac{\sum_{i=1}^{N} p_i \log\left(p_i\right)}{\log\left(N\right)}$$

where $p_i$ is the density the $i$th bin in the histogram, and $N$ is the total number of bins (50, in this case). The resulting entropy can range from 0 to 1, with higher scores indicating less predictable syllable durations, consistent with the songs of immature birds.

Entropy is calculated similarly for silent gap durations, where gap durations are defined as the time difference between a syllable offset and the immediately following syllable onset, up to a maximum duration of 200 ms. A log transform was not applied to gap durations before constructing a histogram with 20 10 ms bins.

### Rhythm spectrograms

Rhythm spectrograms are a visualization of the strength and stereotypy of rhythmic patterns in a bird's song, generated by concatenating the rhythm spectra of multiple song bouts, as first proposed in *Saar and Mitra, 2008*. A song bout's rhythm spectrum is the spectrum of the first derivative of its amplitude. If a song's amplitude has consistent repeating fluctuation patterns (as we expect for a bout composed of multiple repetitions of the same stereotyped motif), then its spectrum will exhibit harmonic banding patterns. If, by contrast, there are no repeating rhythms in the song's amplitude, the rhythm spectrum will have a more even spread of energy across frequency bands. To detect these harmonic patterns more easily in the rhythm spectrogram, all rhythm spectrograms are plotted as the rolling average of 10 song bouts, smoothing out some bout-to-bout variation in the spectra to make harmonic bands more obvious.

To ensure consistent dimensions and resolution across song bouts, the rhythm spectrum is calculated for segments of song of a fixed duration, rather than complete song bouts. This also eliminates the need for any segmentation and labeling of song files to identify bouts, making this timing analysis method completely independent of possible segmentation and labeling errors. Each .wav file is broken into multiple 3-s-long frames, with a hop length of 0.2 s. The three frames with the highest total amplitude (i.e. the three windows containing the most vocalizations) from each file have their

rhythm spectra calculated, and the mean of their spectra is taken as the rhythm spectrum for that file. Because of this windowing system, only files at least 3+3 ×0.2 s in duration can be windowed this way, so shorter .wav files are ignored.

The derivative of the amplitude of each frame is centered at 0 by subtracting the mean value, then multiplied by a Hanning window to reduce spectral leakage when calculating the spectrum. The transformed amplitude derivative is then padded to a total length of 100000 frames, resulting in a smoother spectrum with more interpolated values. A bandpass filter is then applied, keeping only frequency components above 1 Hz and below 500 Hz, as these are the frequencies consistent with typical zebra finch motif and syllable periods. Finally, the real component of the Fourier transform is calculated, constituting the frame's 'rhythm spectrum'. Only portions of the rhythm spectrum corresponding to frequencies between 0 and 30 Hz are included in rhythm spectrograms and downstream feature calculations, as this is the range with the strongest harmonic banding for typical zebra finches. This is all done using AVN's avn.timing.RhythmAnlysis.make_rhythm_spectrogram() function.

### Rhythm spectrum entropy

We quantify the strength of the harmonic content of a bird's rhythm spectrum (i.e. the strength of its rhythm) by calculating the Wiener Entropy of the mean rhythm spectrum across bouts. Wiener entropy is a common acoustic feature used to assess the harmonic nature of zebra finch syllables, with scores near 0 reflecting signals with little harmonic structure, and scores ranging to negative infinity for signals with more harmonic structure.

$$\text{rhythm spectrum entropy} = \frac{\sum_{n=0}^{N} \log\left(\text{rhythm spectrum}^2\right)}{N - \log\left(\sum_{n=0}^{N} \text{rhythm spectrum}^2/N\right)}$$

This is calculated using AVN's avn.timing.RhythmAnalysis.calc_rhythm_spectrogram_entropy() function.

### Peak frequency variability

Whereas the rhythm spectrum entropy measures the overall strength of the rhythms in a set of songs, the peak frequency variability reflects the consistency of the rhythm across multiple song renditions. The exact spacing of the harmonics in a rhythm spectrogram depends on the shape of the amplitude trace of the bird's motif. It isn't obvious how different motifs and motif lengths affect the banding pattern of the rhythm spectrogram, so it doesn't make sense to compare the appearance of different birds' rhythm spectra beyond the prominence of harmonic bands. Likewise, the frequency of the harmonic band with the highest magnitude doesn't carry any special meaning. However, in a very stereotyped bird, that harmonic band will be consistent across songs. If the bird sings its song slightly faster or slower, the band can shift slightly in the frequency domain. So, we measure a bird's rhythm stereotypy by looking at the variability of the frequency with the highest magnitude across song files (the peak frequency).

In practice, the frequency band with the highest energy can jump between harmonic bands across files, even while the overall timing is largely unchanged, so to truly capture fluctuations in the underlying rhythms in a bird's song, we restrict the range of 'peak frequency' values to a 3 Hz band about the median peak frequency across bouts. The CV of the peak frequency within this range is calculated as the peak frequency variability. This is done using AVN's avn.timing.RhythmAnalysis.calc_peak_freq_cv() function.

## Acoustic features

Goodness of pitch, Mean Frequency, Wiener Entropy, Amplitude, Amplitude Modulation, Frequency Modulation, and Pitch were all calculated using AVN in Python, with implementations based on the Sound Analysis Tools for MATLAB (*Tchernichovski et al., 2000*). Each of these features is calculated for each frame in a spectrogram, resulting in a time series of values. We summarize these time series of varying lengths by taking the mean value of each feature for each AVN segmented syllable. We then calculate the mean and coefficient of variation of the mean feature values for each syllable type

according to their AVN labels. As each bird has a different number of syllable types, we need to further summarize these features so that we have a consistent set of values for comparisons across individuals. To do this, we take the syllable type with the minimum, maximum, and median mean value and CV for each feature. This results in six values summarizing the variability and absolute values of each feature for each bird. Across seven acoustic features plus syllable duration, this results in a total set of 48 features.

## Linear discriminant analysis

We fit three different linear discriminant analysis models in this paper. One to discriminate between typical zebra finches and isolate zebra finches, one to discriminate between typical zebra finches and deaf zebra finches, and one to discriminate between all three groups at once. For each of these models, L1 regularization was used to reduce the number of features considered in the model. This improves both the generalization of the model and its interpretability by focusing on just a subset of the most informative features. L1 feature selection was performed considering all AVN features from each bird, excluding amplitude and amplitude-modulation features, as these were found to vary according to recording conditions. Once the feature set was reduced, classification accuracy of the models was tested using a stratified k-fold cross-validation approach. Classification accuracy is the fraction of data points in the withheld test set which were correctly classified. Plotted LDA values and feature weights were obtained from a model trained with the complete dataset. This was all done using the scikit-learn Python package (*Fabian Pedregosa et al., 2011*).

## Age prediction generalized additive model

The full AVN feature set was calculated for 19 individual birds across 103 age points, using songs produced within the first 4 hr after lights on. Juvenile birds have been shown to have more variable songs in the early morning (*Brudner et al., 2023*), which exaggerates the difference between immature and mature song and improves the model's ability to predict a bird's age, compared to features calculated with a full day of songs, or songs produced in the afternoon.

Before fitting a Generalized Additive Model (GAM) for age prediction, we pruned our feature set to include only the most informative features. We first excluded all amplitude and amplitude-modulation features, as these were strongly affected by recording conditions and differed between colonies. We then calculated the mutual information between each remaining feature and age, considering 43 age points from 12 individual birds. We automatically excluded all features with a mutual information score lower than 0.05 (20/44 features). We further refined this feature set by performing forward feature selection with our 43 age point dataset. This means we iteratively added individual features to the model based on which additional feature resulted in the lowest mean squared error (MSE) predictions in a bird-fold cross-validation. We then selected the feature set with the lowest overall MSE, further reducing our feature set to just seven features.

A GAM model with the seven selected features was used to predict bird ages in a leave-one-bird-out (aka bird-fold) cross-validation scheme. Here, we included the 43 age points from 12 individual birds used for feature selection, plus an additional test set of 60 age points from 7 individual birds. We saw no significant difference in model performance between this test set and the dataset used for feature selection, so we pooled results across these groups.

To investigate the contribution of each feature to the overall model, we fit a model with all birds in the dataset and used the pyGAM Python package (*Servén et al., 2018*) to extract the partial dependence functions for each feature.

## Similarity scoring

### Data preparation

Spectrograms of manually segmented and labeled syllables from 21 adult zebra finches from the UTSW colony were used for model training. All validation was performed with spectrograms of WhisperSeg segmented syllables from a test set of 30 tutor-pupil pairs from UTSW and 25 tutor-pupil pairs from the Rockefeller Song Library (*Tchernichovski et al., 2021*), none of which were included in training. For the triplet loss model, these spectrograms were normalized for amplitude, then clipped or padded to a uniform duration of 180 ms. To reduce computational costs, all frequency bands below 2 kHz and above 6 kHz are discarded. For the variational autoencoder model, spectrograms

were computed for each syllable then interpolated to consistent time and frequency representations; 128 frequency bands ranging from 400 Hz to 10 kHz, and 128 time bins. Syllables longer than 300 ms were discarded. Short syllables were stretched in time by a factor of $\sqrt{\frac{300ms}{t}}$, then symmetrically zero-padded to a size of 128 time bins, such that all syllables' spectrograms had the same dimensions.

## Triplet loss model architecture

The proposed neural network model is composed of five convolutional layers alternating with four 'Multiscale Analysis Modules' (MAMs), followed by a global pooling layer and three fully connected linear layers (*Figure 6—figure supplement 3*). This architecture is based on the model proposed in *Thakur et al., 2019*, which was used for species classification with field recordings of bird, frog, and toad vocalizations. The first convolutional layer consists of 32 3x3 kernels, with the 4 subsequent convolutional layers consisting of 64 3x3 kernels with a stride length of 2 along the frequency axis, resulting in downsampling by a factor of 2 along that dimension. Each MAM is composed of four parallel strands, each processing the data at different scales. The first strand consists of a single convolutional layer with 32 1x1 kernels, the second, third, and fourth strands start with a 32 filter 1x1 kernel convolutional layer, followed by a convolutional layer with 32 3x3, 5x5, and 7x7 kernels, respectively. The output of each of these strands is concatenated channel-wise, resulting in a 128-channel representation of the data which is passed to the next layer. This parallel strand organization allows the model to perform feature extraction at multiple different scales without increasing the depth of the model, saving computational cost and limiting potential overfitting. This approach was first proposed in *Szegedy et al., 2015*. The ReLU activation function is used after every layer (*Nair and Hinton, 2010*). The output of the final linear layer is an 8-dimensional vector, which represents an input syllable's embedding. These vectors are normalized to have a length of 1, such that all embeddings lie on a unit eight-dimensional hypersphere.

## Triplet loss model training

The model is trained using dynamic triplet loss with triplet mining. Triplet loss involves presenting the model with batches of triplets, where each triplet consists of an anchor, a positive, and negative syllable. The anchor and positive syllables carry the same manual annotation label from the same bird, and the anchor and negative syllables carry different labels, either from the same bird or from an unrelated bird. The loss function to be minimized is:

$$\text{Loss} = \sum_{i=1}^{N} \max \left( \left|\left| f(A_i) - f(P_i) \right|\right|_2 - \left|\left| f(A_i) - f(N_i) \right|\right|_2 + \alpha, 0 \right)$$

where $N$ is the total number of possible triplets in training, $f(A_i)$ is the embedding of the anchor in triplet $i$, $f(P_i)$ is the embedding of the positive, $f(N_i)$ is the embedding of the negative, and $\alpha$ is a margin parameter. If the positive is closer to the anchor than the negative by at least $\alpha$, the loss for that triplet is 0.

During training, many randomly sampled triplets will already yield a loss of 0, and therefore will not lead to any change in the model. As a result, such triplets are not presented to the model during training. The remaining triplets that result in a positive loss value can be divided into two groups: hard triplets and semi-hard triplets. Hard triplets are cases where $\|f(A) - f(P)\| > \|f(A) - f(N)\|$, and semi-hard triplets are cases where $\|f(A) - f(P)\| < \|f(A) - f(N)\|$ and $\|f(A) - f(P)\| - \|f(A) - f(N)\| > \alpha$. Hard triplets result in higher loss values and, therefore, larger model weight updates, which can result in unstable training. Previous studies have shown that, as a result, training with semi-hard triplets alone can lead to faster and more stable model convergence (*Schroff et al., 2015*). We found that we achieved the best model performance when training with a ratio of 75 semi-hard triplets: 25 hard triplets, so this ratio was used to train the final model. A forward pass of the model is performed for each mini-batch in training to determine the 'hardness' of all possible triplets from that batch. Triplets are then sampled according to the specified ratio of semi-hard to hard triplets and are presented to the model for training.

Our approach is said to be *dynamic* triplet loss because the value of the margin parameter $\alpha$ is updated dynamically over the course of training. The value is initially set to 0.1 and increased stepwise by 0.2 every time a training epoch had fewer than 2500 hard or semi-hard triplets per batch on

average, up to a maximum value of 0.7. This allows the model to begin learning an easier task, of separating syllables of different classes by a smaller margin. As the margin increases, the task gradually becomes more difficult, leading to more stable model convergence compared to starting with a higher margin value. Each time the margin parameter is increased, triplets that previously resulted in a loss of 0 can become semi-hard triplets, meaning the set of triplets presented to the model also expands over the course of training. The initial margin value, maximum margin value, margin step size, and non-zero triplet threshold were all determined empirically. The weight optimization was performed with the Adam optimizer, with weight decay of 0.0001 (*Kingma and Ba, 2014*).

Ultimately, this dynamic triplet loss training constitutes a form of deep metric learning, where high-dimensional inputs (e.g. spectrograms of syllables) are mapped onto a lower-dimensional space where the similarity between samples is proportional to the distance between them. Training with triplets is advantageous in a context where the total amount of labeled training data is limited, as the number of possible triplets is proportional to the cube of the number of training samples.

### VAE

In addition to using this triplet loss model, we also calculated similarity score using syllable embeddings generated by a variational autoencoder (VAE) model, as described in *Goffinet et al., 2021*. Briefly, using the *autoencoded-vocal-analysis* Python package (*Goffinet et al., 2021*), we trained the base VAE model with an eight-dimensional latent embedding using syllable spectrograms from the 21 UTSW zebra finches used to train the triplet loss model.

### EMD

Syllable embeddings were obtained by running a forward pass of the trained triplet loss or VAE model with all segmented syllables that an individual bird produced on a given day. This results in a set of thousands of syllables in the 8-dimensional embedding space. Two bird songs are compared by calculating the Earth Mover's Distance (EMD) between their syllable embeddings using the PyEMD package (*Doran, 2014*). If one were to imagine the syllable embedding empirical distributions as piles of dirt, the earth mover's distance is the minimum cost of moving the earth from one distribution to match the other, where cost is defined as the amount of dirt moved multiplied by the distance over which it is moved. This value can range from 0 for identical distributions to positive infinity for distributions that are infinitely far apart. As our triplet loss model's embedding space is limited to points lying on a unit radius 8-dimensional hypersphere, the maximum possible value for EMD is 1.41 for distributions that are each concentrated on a single point, maximally separated within the constraints of the embedding space.

The EMD score considers many song renditions from each bird being compared, allowing a better overall comparison of the similarity between two birds' song production as compared to multiple pairwise comparisons between renditions. One limitation of EMD, however, is that it is completely agnostic to syllable sequencing, so a pupil that imitated all syllables from his tutor but sings them in a completely different order will have a similar EMD score to a pupil that imitated all syllables from a tutor and produces them in the same order. That said, there is new evidence that zebra finches recognize songs independently of syllable order, raising questions about the importance of syllable order in song perception (*Ning et al., 2023*). The EMD score is also symmetrical, meaning that the presence of syllables in the tutor's song that weren't imitated by the pupil will have the same impact on the EMD score as new, improvised syllables present in the pupil's song but not in the tutor's song.

### MMD

In addition to EMD, we calculate the dissimilarity between distributions of embedded syllables using maximum mean discrepancy (MMD), as in *Goffinet et al., 2021*. MMD compares two distributions of syllables by computing the squared distance between their mean features mapped in a reproducing kernel Hilbert space (RKHS) using a Gaussian radial basis function (RBF) kernel, with a kernel bandwidth equal to half the median pairwise distance between 1000 randomly sampled syllable embeddings from our validation dataset. The result is a measure of the dissimilarity between distributions, where a larger score indicates greater dissimilarity between the two syllable distributions. As with EMD, this is a symmetrical score that doesn't consider the syllable sequencing similarity between the two birds being compared.

## Similarity scores across comparison types

To validate the performance of our models and EMD or MMD scores for similarity scoring, we compute EMD and MMD scores between a pupil and itself, a pupil and its tutor, pupil and a 'sibling', and a pupil and an unrelated bird. For comparisons between a pupil and itself, embeddings of all recorded syllables from a day of song are computed. If the bird has fewer than 4000 syllables in this dataset, the dataset is randomly split in half and the two halves are compared. If a bird has more than 4000 syllables, two sets of 2000 syllables are randomly sampled and compared. For comparisons between a pupil and its tutor, up to 4000 syllables are sampled from each of the tutor and the pupil and compared. In the case of pupil and 'sibling' comparisons, a sibling is defined as another bird sharing the same song tutor. We expect that typical zebra finches that learned from the same tutor will have similar songs, but that these will generally be less similar than a pupil compared to its tutor directly. Each pupil is compared to up to three 'siblings', depending on availability in our dataset. Finally, each pupil is compared to three randomly selected pupils who don't share their song tutor. For each of these comparisons, up to 4000 syllables are randomly sampled from each bird as well, to help reduce the compute time for EMD and MMD.

## Contrast index

As in **Mandelblat-Cerf and Fee, 2014**, contrast index is calculated as:

$$\text{Contrast Index} = \frac{\text{self similarity - cross similarity}}{\text{self similarity + cross similarity}}$$

where *self-similarity* is the EMD or MMD score between pupil and itself, calculated as described in the previous section, and *cross similarity* is the mean similarity between a pupil and 3 unrelated birds, again calculated as described in the previous section. As EMD and MMD are *dissimilarity* scores, rather than similarity scores, more negative values reflect better contrast between comparisons, so we report the absolute value for ease of comparison to existing similarity scoring methods.

## Tutor contrast index

Similarly to the contrast index score, the tutor contrast index is calculated as:

$$\text{Tutor Contrast Index} = \frac{\text{tutor similarity} - \text{cross similarity}}{\text{tutor similarity} + \text{cross similarity}}$$

where *tutor similarity* is the EMD or MMD score between a pupil and its tutor, calculated as described previously, and *cross similarity* is the mean similarity between a pupil and three unrelated birds. As with the contrast index, absolute values are reported, with higher values indicating larger contrast in score between a pupil vs. tutor comparison and pupil vs. unrelated bird comparison for a given pupil bird.

## Expert human similarity scores

A panel of 11 expert human annotators was each presented with 126 pairs of spectrograms and was instructed to rate their similarity on a scale from 1 (not similar) to 10 (very similar). The spectrograms were generated using Sound Analysis Pro 2011 (**Tchernichovski et al., 2000**), began at the beginning of a song bout, and included at least one full motif when motif structure was present. Raters were presented with 4 pupil-tutor spectrogram pairs per pupil, 8 comparisons between a tutor and itself to ensure that raters were using the full rating scale, and 10 duplicated tutor-pupil comparisons to ensure that the scorers were internally consistent. No individual scorer differed from the mean score by more than an average of 2 standard deviations, none differed by more than 2 points on the duplicated comparisons, and all but one made use of the full scale (this scorer never assigned a perfect 10/10 score, so their scores were rescaled such that they spanned the full range). Similarity scores for each tutor-pupil pair were obtained by taking the mean similarity score across their 4 spectrogram pairs, across all scorers. The full scoring set is available at https://forms.gle/9TDu1fwGGYXWKhgB6. These scores were previously generated for and published in **Garcia-Oscos et al., 2021**. Of the birds evaluated, 15 were also in the similarity scoring validation set, so the correlation between these 15

birds' mean human similarity scores and tutor-pupil EMD and MMD scores was used to evaluate the agreement between methods.

For comparison to the expert human similarity scores, Sound Analysis Pro 2011 % similarity scores were calculated for the same set of 15 pupils. A representative motif from the tutor song was selected and compared to between 30 and 60 motif renditions from the pupil bird when the pupil was over 90 dph, using the asymmetric time-courses similarity tool. The final reported scores are the mean % similarity across all comparisons for a given pupil.

### Comparisons to mature song

For the six birds from UTSW and five birds from Duke University (*Brudner et al., 2023*) from which we had recordings at 90-100d ph and earlier time points, we computed the MMD between their juvenile and adult songs. For each bird, 4000 WhisperSeg segmented syllables were sampled from a full day of song recordings when the birds were between 90 and 100 dph, to serve as the mature song distribution. Up to 4000 WhisperSeg segmented syllables were sampled from each day of available recordings prior to or shortly following the mature song date for comparison. MMD scores were calculated using embeddings from the triplet loss model as described previously. As the scores appeared to follow an exponential pattern, where the rate of song dissimilarity change slowed over development, we fit an exponential function to the data using the scipy.optimize curve_fit() function (*Virtanen et al., 2020*), and plotted this function alongside the data.

## Acknowledgements

We thank Fayha Zia for help with manual syllable labeling, Drs. Chihito Mori and Kazuhiro Wada for sharing recordings of early-deafened zebra finches, Drs.Ofer Tchernichovski and Erich Jarvis for publicly sharing the Rockefeller University Field Research Center Song Library, and Dr. Richard Mooney for publicly sharing recordings of juvenile zebra finches across development. We thank Drs. Tyler Lee and Daisuke Hattori for their valuable feedback and suggestions on AVN's design and validation. We also thank Luis Garcia, Andrea Guerrero, and all members of the Roberts Lab for bird care support and their generous feedback on this work. This research was supported by the US National Institutes of Health R01 DC020333 to TFR. TMIK was supported by a Neural Scientist Training Program Fellowship from the UT Southwestern O'Donnell Brain Institute. ESM was supported by the UTD/UTSW Green Fellowship.

## Additional information

### Funding

| Funder | Grant reference number | Author |
| --- | --- | --- |
| National Institutes of Health | R01 DC020333 | Todd F Roberts |
| The University of Texas Southwestern Medical Center | Neural Scientist Training Program | Therese MI Koch |
| The University of Texas at Dallas | UTSW Green Fellow's Program | Ethan S Marks |

The funders had no role in study design, data collection and interpretation, or the decision to submit the work for publication.

### Author contributions

Therese MI Koch, Conceptualization, Data curation, Software, Formal analysis, Funding acquisition, Validation, Investigation, Visualization, Methodology, Writing – original draft, Writing – review and editing; Ethan S Marks, Software, Validation, Visualization, Methodology, Writing – review and editing; Todd F Roberts, Conceptualization, Supervision, Funding acquisition, Project administration, Writing – review and editing

**Author ORCIDs**
Therese MI Koch  https://orcid.org/0000-0002-5327-3219
Ethan S Marks  https://orcid.org/0009-0004-4415-9889
Todd F Roberts  https://orcid.org/0000-0002-0967-6598

**Ethics**
Experiments described in this study were conducted using male zebra finches. All procedures were performed in accordance with protocols approved by Animal Care and Use Committee at UT Southwestern Medical Center, protocol number 2016-101562-G.

Reviewer #2 (Public review): https://doi.org/10.7554/eLife.101111.3.sa1
Reviewer #3 (Public review): https://doi.org/10.7554/eLife.101111.3.sa2
Author response https://doi.org/10.7554/eLife.101111.3.sa3

---

## Additional files

### Supplementary files
MDAR checklist
Supplementary file 1. Subject bird list.

### Data availability
Song recordings and annotations for all birds recorded at UTSW are available through the Texas Data Repository (https://dataverse.tdl.org/dataverse/avn). Song recordings from the Rockefeller University Song Library are available at http://ofer.hunter.cuny.edu/songs. Annotations of songs from select birds in the Rockefeller Song Library were generated for this publication and are available through the Texas Data Repository (https://doi.org/10.18738/T8/DN0SIV). Recordings of juvenile birds from Duke University are available at https://doi.org/10.7924/r4j38x43h. Recordings of early-deafened zebra finches are available upon request from Dr. Kazuhiro Wada. The AVN documentation, AVN-GUI, and code necessary to produce all figures in this manuscript can be found through the following links: AVN documentation: https://avn.readthedocs.io/en/latest/index.html; AVN-GUI: https://avn.readthedocs.io/en/latest/AVN_GUI.html; Code for figures: https://github.com/theresekoch/AVN_paper (copy archived at *Koch, 2025a*); Model training: https://github.com/theresekoch/Triplet_loss_model_training (*Koch, 2025b*).

The following datasets were generated:

| Author(s) | Year | Dataset title | Dataset URL | Database and Identifier |
|-----------|------|---------------|-------------|-------------------------|
| Koch TMI | 2024 | Labeled Zebra Finch Songs | https://doi.org/10.18738/T8/SAWMUN | Texas Data Repository, 10.18738/T8/SAWMUN |
| Koch TMI | 2024 | Labels for Rockefeller Song Library Birds | https://doi.org/10.18738/T8/DN0SIV | Texas Data Repository, 10.18738/T8/DN0SIV |

The following previously published dataset was used:

| Author(s) | Year | Dataset title | Dataset URL | Database and Identifier |
|-----------|------|---------------|-------------|-------------------------|
| Brudner S, Pearson J, Mooney R | 2022 | Data from: Juvenile zebra finch syllables for data-driven analysis of development | https://doi.org/10.7924/r4j38x43h | Duke Research Data Repository, 10.7924/r4j38x43h |

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
