## [Editor Report · eLife Assessment]

This work introduces a new Python package, Avian Vocalization Analysis (AVN) that provides several key analysis pipelines for birdsong research. This tool is likely to prove **useful** to researchers in neuroscience and beyond, as demonstrated by **convincing** experiments using a wide range of publicly available birdsong data.

---

## [Referee Report · Reviewer #2 (Public review)]

Summary:

In this work, the authors present a new Python software package, Avian Vocalization Network (AVN) aimed at facilitating the analysis of birdsong, especially the song of the zebra finch, the most common songbird model in neuroscience. The package handles some of the most common (and some more advanced) song analyses, including segmentation, syllable classification, featurization of song, calculation of tutor-pupil similarity, and age prediction, with a view toward making the entire process friendlier to experimentalists with limited coding experience working in the field.

For many years, Sound Analysis Pro has served as a standard in the songbird field, the first package to extensively automate songbird analysis and facilitate the computation of acoustic features that have helped define the field. More recently, the increasing popularity of Python as a language, along with the emergence of new machine learning methods, has resulted in a number of new software tools, including the vocalpy ecosystem for audio processing, TweetyNet (for segmentation), t-SNE and UMAP (for visualization), and autoencoder-based approaches for embedding.

As with any software package, this one necessarily makes a number of design choices, which may or may not fit the needs of all users. Those who prefer a more automated pipeline with fewer knobs to turn may appreciate AVN in cases where the existing recipes fit their needs, while those who require more customization and flexibility may require a more bespoke (and thus code-intensive) approach.

Strengths:

The AVN package overlaps several of these earlier efforts, albeit with a focus on more traditional featurization that many experimentalists may find more interpretable than deep learning-based approaches. Among the strengths of the paper are its clarity in explaining the several analyses it facilitates, along with high-quality experiments across multiple public datasets collected from different research groups. As a software package, it is open source, installable via the pip Python package manager, and features high-quality documentation, as well as tutorials. For experimentalists who wish to replicate any of the analyses from the paper, the package is likely to be a useful time saver.

Weaknesses:

I think the potential limitations of the work are predominantly on the software end, with one or two quibbles about the methods.

First, the software: It's important to note that the package is trying to do many things, of which it is likely to do several well and a few comprehensively. Rather than a package that presents a number of new analyses or a new analysis framework, it is more a codification of recipes, some of which are reimplementations of existing work (SAP features), some of which are essentially wrappers around other work (interfacing with WhisperSeg segmentations), and some of which are new (similarity scoring). All of this has value, but in my estimation, it has less value as part of a standalone package and potentially much more as part of an ecosystem like vocalpy that is undergoing continuous development and has long-term support. While the code is well-documented, including web-based documentation for both the core package and the GUI, the latter is available only on Windows, which might limit the scope of adoption.

That is to say, whether AVN is adopted by the field in the medium term will have much more to do with the quality of its maintenance and responsiveness to users than any particular feature, but I believe that many of the analysis recipes that the authors have carefully worked out may find their way into other code and workflows.

In the revised version of the paper, the authors have expanded their case for the design choices made in AVN and remain committed to maintaining the tool. Given the low cost for users in trying new methods and the work the authors have put into further reducing this overhead via documentation, those curious about the package are likely best served by simply downloading it and giving it a try on their own data.

Second, two notes about new analysis approaches:

(1) The authors propose a new means of measuring tutor-pupil similarity based on first learning a latent space of syllables via a self-supervised learning (SSL) scheme and then using the earth mover's distance (EMD) to calculate transport costs between the distributions of tutors' and pupils' syllables. While, to my knowledge, this exact method has not previously been proposed in birdsong, I suspect it is unlikely to differ substantially from the approach of autoencoding followed by MMD used in the Goffinet et al. paper. That is, SSL, like the autoencoder, is a latent space learning approach, and EMD, like MMD, is an integral probability metric that measures discrepancies between two distributions. (Indeed, the two are very closely related: https://stats.stackexchange.com/questions/400180/earth-movers-distance-and-maximum-mean-discrepency) Without further experiments, it is hard to tell whether these two approaches differ meaningfully. Likewise, while the authors have trained on a large corpus of syllables to define their latent space in a way that generalizes to new birds, it is unclear why such an approach would not work with other latent space learning methods.

Update: The authors now provide an extensive comparison with the Goffinet et al. paper and also consider differences between MMD and EMD. This comparison both adds value to the original paper and provides useful benchmarking for others looking to develop latent space comparison methods.

(2) The authors propose a new method for maturity scoring by training a model (a generalized additive model) to predict the age of the bird based on a selected subset of acoustic features. This is distinct from the "predicted age" approach of Brudner, Pearson, and Mooney, which predicts based on a latent representation rather than specific features, and the GAM nicely segregates the contribution of each. As such, this approach may be preferred by many users who appreciate its interpretability.

In summary, my view is that this is a nice paper detailing a well-executed piece of software whose future impact will be determined by the degree of support and maintenance it receives from others over the near and medium term.

---

## [Referee Report · Reviewer #3 (Public review)]

This paper introduces the Avian Vocalization Network (AVN), a novel birdsong analysis pipeline using deep learning. By automating vocal annotation tasks, the AVN generates interpretable song features and song similarity scores on novel datasets without retraining. The performance of the network is solid and is comparable to that of human annotators.

The authors have improved the manuscript in several aspects, such as the comparison with the Goffinet work. Overall, the AVN feature set could become a useful tool for evaluating birdsongs. But the authors also chose not to address a certain number of criticisms, and some issues remain poorly addressed, and the work is not reproducible at this stage. With a little effort, these issues could get resolved in my view. I will just pick on four issues that I think can be easily addressed:

(1) Limitation of feature set: They claim that AVN satisfies the criteria (line 60) of "creating a common feature space for the comparison of behavioural phenotypes ..."(line 51), but then on LDA analysis, explained on line 910 they say "excluding amplitude and amplitude modulation features as they were found to vary". Since their feature set is not stable and not truly 'common' to all tasks, this limitation needs addressing in the discussion (that some features seem to vary undesirably, and they need exclusion based on some criteria to be defined).

(2) Missing information on classification training loss: The Authors insist that their triplet loss is not related to classification, and they brush off my request for more information. In their rebuttal, they write: 'The loss function is related to the relative distance between embeddings of syllables with the same or different labels, not the classification of syllables as same or different.' Perplexingly, however, in the revised paper, authors speak themselves of 'classes', in Line 1004: this allows the model to begin learning an easier task, of separating syllables of different classes by a smaller margin.' So it seems the authors actually agree with me that there is an underlying classification task. I am therefore going to make it a bit more explicit here what I'm asking for, hoping this will better resonate with them.

In line 984 they define their loss function and in lines 994-996 they define 'hard' and 'semi-hard' triplets. Authors then train a system to minimize the loss with a ratio of 75 percent semi-hard triplets and 25 percent hard triplets and a final weighing parameter value alpha=0.7. What I'm asking for is this 'classification' loss their trained model achieves, or in other words, the fraction of triplets that end up producing a loss, either of the 'hard' or 'semi-hard' type. For example, if their model manages to separate all 'possible triplets' by a margin of at least alpha, then the loss would be zero. If the model achieves to separate all triplets except one, then the loss would correspond to the amount by which the separation differences between the anchor and the positive vs negative samples exceeds alpha. So, an important number to provide in the paper is the fraction of triplets that incur a nonzero loss, i.e., the fraction of semi-hard triplets. And another important quantity is the fraction of hard triplets, i.e. the fraction of triplets that would incur a loss if alpha were set to zero, or, in other words, the triplets for which the negative sample is closer to the anchor than the positive sample. By the way, I assume this latter fraction of hard cases will be zero - that their model does not confuse any positive and negative training samples...

Note: the quantification chosen by the authors termed 'contrast index' is interesting, but it is a derived quantity, it is not the quantity authors chose to optimize during training. If authors were to report both the training loss achieved and the 'contrast index', follow-up work could be benchmarked against both these quantities. If for example, a follow-up model achieves smaller loss but worse contrast, then the loss is not a good placeholder measure for optimizing contrast. Alternatively, follow-up work could focus on the contrast index as training objective, obliterating the need for the triplet loss as an intermediate step (I don't buy the authors' argument that such an optimization would be infeasible).

(3) Reproducibility: they explain the way they train the CNN with triplet loss to produce the embeddings, but we're missing both actual scripts on GitHub to train and inference from scratch, and model weights, or even hyper parameters they used. Authors only provide the architecture, and I don't think that's enough to be considered replicable in today's standards. I would suggest they release complete model checkpoint weights for the result they report, the exact data splits, the hyper parameters they used and training and testing code, so that one can very easily verify their claims and apply their methods to other datasets. Note: for example, the code to extract the embeddings is incomplete (the function definition of single_bird_extract_embeddings cannot be found on GitHub) and the model weights they used are missing.

(4) With regards to the age prediction model, the authors should specify that this model is mainly useful for comparisons across studies but less so for precise evaluation of the effects of a treatment within a study. Namely, the effect on song of a treatment is best assessed by comparison to within-subject past song, and by comparison to age-matched control birds (ideally siblings) raised in identical conditions, rather than to invoke a generic model trained on other birds and from different colonies and breeding conditions as authors propose to do. In other words, to introduce a generic model for evaluation of song maturity introduces measurement noise in terms of the additional birds and their variable conditions, which can hinder precise assessment of treatment effects. Note that to state that in past work such maturity models were used is not a good justification, scientifically speaking.

Finally, the authors write that methods for syllable segmentation have not been systematically compared but the whisperseg work they use did such a comparison. So the authors should revise their novelty claim of being the first to compare syllable segmentation methods.

---

## [Author Response]

The following is the authors’ response to the original reviews.

**Reviewer #1 (Public Review):**
Summary:This paper applies methods for segmentation, annotation, and visualization of acoustic analysis to zebra finch song. The paper shows that these methods can be used to predict the stage of song development and to quantify acoustic similarity. The methods are solid and are likely to provide a useful tool for scientists aiming to label large datasets of zebra finch vocalizations. The paper has two main parts: (1) establishing a pipeline/ package for analyzing zebra finch birdsong and (2) a method for measuring song imitation.Strengths:It is useful to see existing methods for syllable segmentation compared to new datasets.It is useful, but not surprising, that these methods can be used to predict developmental stage, which is strongly associated with syllable temporal structure.It is useful to confirm that these methods can identify abnormalities in deafened and isolated songs.Weaknesses:For the first part, the implementation seems to be a wrapper on existing techniques. For instance, the first section talks about syllable segmentation; they made a comparison between whisperseg (Gu et al, 2024), tweetynet (Cohen et al, 2022), and amplitude thresholding. They found that whisperseg performed the best, and they included it in the pipeline. They then used whisperseg to analyze syllable duration distributions and rhythm of birds of different ages and confirmed past findings on this developmental process (e.g. Aronov et al, 2011). Next, based on the segmentation, they assign labels by performing UMAP and HDBScan on the spectrogram (nothing new; that's what people have been doing). Then, based on the labels, they claimed they developed a 'new' visualization - syntax raster (line 180). That was done by Sainburg et. al. 2020 in Figure 12E and also in Cohen et al, 2020 - so the claim to have developed 'a new song syntax visualization' is confusing. The rest of the paper is about analyzing the finch data based on AVN features (which are essentially acoustic features already in the classic literature).

First, we would like to thank this reviewer for their kind comments and feedback on this manuscript. It is true that many of the components of this song analysis pipeline are not entirely novel in isolation. Our real contribution here is bringing them together in a way that allows other researchers to seamlessly apply automated syllable segmentation, clustering, and downstream analyses to their data. That said, our approach to training TweetyNet for syllable segmentation is novel. We trained TweetyNet to recognize vocalizations vs. silence across multiple birds, such that it can generalize to new individual birds, whereas Tweetynet had only ever been used to annotate song syllables from birds included in its training set previously. Our validation of TweetyNet and WhisperSeg in combination with UMAP and HDBSCAN clustering is also novel, providing valuable information about how these systems interact, and how reliable the completely automatically generated labels are for downstream analysis. We have added a couple sentences to the introduction to emphasize the novelty of this approach and validation.

Our syntax raster visualization does resemble Figure 12E in Sainburg et al. 2020, however it differs in a few important ways, which we believe warrant its consideration as a novel visualization method. First, Sainburg et al. represent the labels across bouts in real time; their position along the x axis reflects the time at which each syllable is produced relative to the start of the bout. By contrast, our visualization considers only the index of syllables within a bout (ie. First syllable vs. second syllable etc) without consideration of the true durations of each syllable or the silent gaps between them. This makes it much easier to detect syntax patterns across bouts, as the added variability of syllable timing is removed. Considering only the sequence of syllables rather than their timing also allows us to more easily align bouts according to the first syllable of a motif, further emphasizing the presence or absence of repeating syllable sequences without interference from the more variable introductory notes at the start of a motif. Finally, instead of plotting all bouts in the order in which they were produced, our visualization orders bouts such that bouts with the same sequence of syllables will be plotted together, which again serves to emphasize the most common syllable sequences that the bird produces. These additional processing steps mean that our syntax raster plot has much starker contrast between birds with stereotyped syntax and birds with more variable syntax, as compared to the more minimally processed visualization in Sainburg et al. 2020. There doesn’t appear to be any similar visualizations in Cohen et al. 2020.

The second part may be something new, but there are opportunities to improve the benchmarking. It is about the pupil-tutor imitation analysis. They introduce a convolutional neural network that takes triplets as an input (each tripled is essentially 3 images stacked together such that you have anchor, positive, negative), Anchor is a reference spectrogram from, say finch A; positive means a different spectrogram with the same label as anchor from finch A, and negative means a spectrogram not related to A or different syllable label from A. The network is then trained to produce a low-dimensional embedding by ensuring the embedding distance between anchor and positive is less than anchor and negative by a certain margin. Based on the embedding, they then made use of earth mover distance to quantify the similarity in the syllable distribution among finches. They then compared their approach performance with that of sound analysis pro (SAP) and a variant of SAP. A more natural comparison, which they didn't include, is with the VAE approach by Goffinet et al. In this paper (https://doi.org/10.7554/eLife.67855, Fig 7), they also attempted to perform an analysis on the tutor pupil song.

We thank the reviewer for this suggestion. We have included a comparison of our triplet loss embedding model to the VAE model proposed in Goffinet et al. 2021. We also included comparisons of similarity scoring using each of these embedding models combined with either earth mover’s distance (EMD) or maximum mean discrepancy (MMD) to calculate the similarity of the embeddings, as was done in Goffinet et al. 2021. As discussed in the updated results section of the paper and shown in the new Figure 6–figure supplement 1, the Triplet loss model with MMD performs best for evaluating song learning on new birds, not included in model training. We’ve updated the main text of the paper to reflect this switch from EMD to MMD for the primary similarity scoring approach.

Reviewer #2 (Public Review):Summary:In this work, the authors present a new Python software package, Avian Vocalization Network (AVN) aimed at facilitating the analysis of birdsong, especially the song of the zebra finch, the most common songbird model in neuroscience. The package handles some of the most common (and some more advanced) song analyses, including segmentation, syllable classification, featurization of song, calculation of tutor-pupil similarity, and age prediction, with a view toward making the entire process friendlier to experimentalists working in the field.For many years, Sound Analysis Pro has served as a standard in the songbird field, the first package to extensively automate songbird analysis and facilitate the computation of acoustic features that have helped define the field. More recently, the increasing popularity of Python as a language, along with the emergence of new machine learning methods, has resulted in a number of new software tools, including the vocalpy ecosystem for audio processing, TweetyNet (for segmentation), t-SNE and UMAP (for visualization), and autoencoder-based approaches for embedding.Strengths:The AVN package overlaps several of these earlier efforts, albeit with a focus on more traditional featurization that many experimentalists may find more interpretable than deep learning-based approaches. Among the strengths of the paper are its clarity in explaining the several analyses it facilitates, along with high-quality experiments across multiple public datasets collected from different research groups. As a software package, it is open source, installable via the pip Python package manager, and features high-quality documentation, as well as tutorials. For experimentalists who wish to replicate any of the analyses from the paper, the package is likely to be a useful time saver.Weaknesses:I think the potential limitations of the work are predominantly on the software end, with one or two quibbles about the methods.First, the software: it's important to note that the package is trying to do many things, of which it is likely to do several well and few comprehensively. Rather than a package that presents a number of new analyses or a new analysis framework, it is more a codification of recipes, some of which are reimplementations of existing work (SAP features), some of which are essentially wrappers around other work (interfacing with WhisperSeg segmentations), and some of which are new (similarity scoring). All of this has value, but in my estimation, it has less value as part of a standalone package and potentially much more as part of an ecosystem like vocalpy that is undergoing continuous development and has long-term support.

We appreciate this reviewer’s comments and concerns about the structure of the AVN package and its long-term maintenance. We have considered incorporating AVN into the VocalPy ecosystem but have chosen not to for a few key reasons. (1) AVN was designed with ease of use for experimenters with limited coding experience top of mind. VocalPy provides excellent resources for researchers with some familiarity with object-oriented programming to manage and analyze their datasets; however, we believe it may be challenging for users without such experience to adopt VocalPy quickly. AVN’s ‘recipe’ approach, as you put it, is very easily accessible to new users, and allows users with intermediate coding experience to easily navigate the source code to gain a deeper understanding of the methodology. AVN also consistently outputs processed data in familiar formats (tables in .csv files which can be opened in excel), in an effort to make it more accessible to new users, something which would be challenging to reconcile with VocalPy’s emphasis on their `dataset`classes. (2) AVN and VocalPy differ in their underlying goals and philosophies when it comes to flexibility vs. standardization of analysis pipelines. VocalPy is designed to facilitate mixing-and-matching of different spectrogram generation, segmentation, annotation etc. approaches, so that researchers can design and implement their own custom analysis pipelines. This flexibility is useful in many cases. For instance, it could allow researchers who have very different noise filtering and annotation needs, like those working with field recordings versus acoustic chamber recordings, to analyze their data using this platform. However, when it comes to comparisons across zebra finch research labs, this flexibility comes at the expense of direct comparison and integration of song features across research groups. This is the context in which AVN is most useful. It presents a single approach to song segmentation, labeling, and featurization that has been shown to generalize well across research groups, and which allows direct comparisons of the resulting features. AVN’s single, extensively validated, standard pipeline approach is fundamentally incompatible with VocalPy’s emphasis on flexibility. We are excited to see how VocalPy continues to evolve in the future, and recognize the value that both AVN and VocalPy bring to the songbird research community, each with their own distinct strengths, weaknesses, and ideal use cases.

While the code is well-documented, including web-based documentation for both the core package and the GUI, the latter is available only on Windows, which might limit the scope of adoption.

We thank the reviewer for their kind words about AVN’s documentation. We recognize that the GUI’s exclusive availability on Windows is a limitation, and we would be happy to collaborate with other researchers and developers in the future to build a Mac compatible version, should the demand present itself. That said, the python package works on all operating systems, so non-Windows users still have the ability to use AVN that way.

That is to say, whether AVN is adopted by the field in the medium term will have much more to do with the quality of its maintenance and responsiveness to users than any particular feature, but I believe that many of the analysis recipes that the authors have carefully worked out may find their way into other code and workflows.Second, two notes about new analysis approaches:(1) The authors propose a new means of measuring tutor-pupil similarity based on first learning a latent space of syllables via a self-supervised learning (SSL) scheme and then using the earth mover's distance (EMD) to calculate transport costs between the distributions of tutors' and pupils' syllables. While to my knowledge this exact method has not previously been proposed in birdsong, I suspect it is unlikely to differ substantially from the approach of autoencoding followed by MMD used in the Goffinet et al. paper. That is, SSL, like the autoencoder, is a latent space learning approach, and EMD, like MMD, is an integral probability metric that measures discrepancies between two distributions. (Indeed, the two are very closely related: https://stats.stackexchange.com/questions/400180/earth-movers-distance-and-maximum-mean-discrepency) Without further experiments, it is hard to tell whether these two approaches differ meaningfully. Likewise, while the authors have trained on a large corpus of syllables to define their latent space in a way that generalizes to new birds, it is unclear why such an approach would not work with other latent space learning methods.

We recognize the similarities between these approaches and have included comparisons of the VAE and MMD as in the Goffinet paper to our triplet loss model and EMD. As discussed in the updated results section of the paper and shown in the new Figure 6–figure supplement 1, the Triplet loss model with MMD performs best for evaluating song learning on new birds, not included in model training. We’ve updated the main text of the paper to reflect this switch from EMD to MMD for the primary similarity scoring approach.

(2) The authors propose a new method for maturity scoring by training a model (a generalized additive model) to predict the age of the bird based on a selected subset of acoustic features. This is distinct from the "predicted age" approach of Brudner, Pearson, and Mooney, which predicts based on a latent representation rather than specific features, and the GAM nicely segregates the contribution of each. As such, this approach may be preferred by many users who appreciate its interpretability.In summary, my view is that this is a nice paper detailing a well-executed piece of software whose future impact will be determined by the degree of support and maintenance it receives from others over the near and medium term.
**Reviewer #3 (Public Review):**
Summary:The authors invent song and syllable discrimination tasks they use to train deep networks. These networks they then use as a basis for routine song analysis and song evaluation tasks. For the analysis, they consider both data from their own colony and from another colony the network has not seen during training. They validate the analysis scores of the network against expert human annotators, achieving a correlation of 80-90%.Strengths:(1) Robust Validation and Generalizability: The authors demonstrate a good performance of the AVN across various datasets, including individuals exhibiting deviant behavior. This extensive validation underscores the system's usefulness and broad applicability to zebra finch song analysis, establishing it as a potentially valuable tool for researchers in the field.(2) Comprehensive and Standardized Feature Analysis: AVN integrates a comprehensive set of interpretable features commonly used in the study of bird songs. By standardizing the feature extraction method, the AVN facilitates comparative research, allowing for consistent interpretation and comparison of vocal behavior across studies.(3) Automation and Ease of Use. By being fully automated, the method is straightforward to apply and should introduce barely an adoption threshold to other labs.(4) Human experts were recruited to perform extensive annotations (of vocal segments and of song similarity scores). These annotations released as public datasets are potentially very valuable.Weaknesses:(1) Poorly motivated tasks. The approach is poorly motivated and many assumptions come across as arbitrary. For example, the authors implicitly assume that the task of birdsong comparison is best achieved by a system that optimally discriminates between typical, deaf, and isolated songs. Similarly, the authors assume that song development is best tracked using a system that optimally estimates the age of a bird given its song. My issue is that these are fake tasks since clearly, researchers will know whether a bird is an isolated or a deaf bird, and they will also know the age of a bird, so no machine learning is needed to solve these tasks. Yet, the authors imagine that solving these placeholder tasks will somehow help with measuring important aspects of vocal behavior.

We appreciate this reviewer’s concerns and apologize for not providing sufficiently clear rationale for the inclusion of our phenotype classifier and age regression models in the original manuscript. These tasks are not intended to be taken as a final, ultimate culmination of the AVN pipeline. Rather, we consider the carefully engineered 55-interpretable feature set to be AVN’s final output, and these analyses serve merely as examples of how that feature set can be applied. That said, each of these models do have valid experimental use cases that we believe are important and would like to bring to the attention of the reviewer.

For one, we showed how the LDA model that can discriminate between typical, deaf, and isolate birds’ songs not only allows us to evaluate which features are most important for discriminating between these groups, but also allows comparison of the FoxP1 knock-down (FP1 KD) birds to each of these phenotypes. Based on previous work (Garcia-Oscos et al. 2021), we hypothesized that FP1 KD in these birds specifically impaired tutor song memory formation while sparing a bird’s ability to refine their own vocalizations through auditory feedback. Thus, we would expect their songs to resemble those of isolate birds, who lack a tutor song memory, but not to resemble deaf birds who lack a tutor song memory and auditory feedback of their own vocalizations to guide learning. The LDA model allowed us to make this comparison quantitatively for the first time and confirm our hypothesis that FP1 KD birds’ songs are indeed most like isolates’. In the future, as more research groups publish their birds’ AVN feature sets, we hope to be able to make even more fine-grained comparisons between different groups of birds, either using LDA or other similar interpretable classifiers.

The age prediction model also has valid real-world use cases. For instance, one might imagine an experimental manipulation that is hypothesized to accelerate or slow song maturation in juvenile birds. This age prediction model could be applied to the AVN feature sets of birds having undergone such a manipulation to determine whether their predicted ages systematically lead or lag their true biological ages, and which song features are most responsible for this difference. We didn’t have access to data for any such birds for inclusion in this paper, but we hope that others in the future will be able to take inspiration from our methodology and use this or a similar age regression model with AVN features in their research. We have added a couple lines to the ‘Comparing Song Disruptions with AVN Features’ and ‘Tracking Song Development with AVN Features’ sections of the results to make this more clear.

Along similar lines, authors assume that a good measure of similarity is one that optimally performs repeated syllable detection (i.e. to discriminate same syllable pairs from different pairs). The authors need to explain why they think these placeholder tasks are good and why no better task can be defined that more closely captures what researchers want to measure. Note: the standard tasks for self-supervised learning are next word or masked word prediction, why are these not used here?

This reviewer appears to have misunderstood our similarity scoring embedding model and our rationale for using it. We will explain it in more depth here and have added a paragraph to the ‘Measuring Song Imitation’ section of the results explaining this rationale more briefly.

First, nowhere are we training a model to discriminate between same and different syllable pairs. The triplet loss network is trained to embed syllables in an 8-dimensional space such that syllables with the same label are closer together than syllables with different labels. The loss function is related to the relative distance between embeddings of syllables with the same or different labels, not the classification of syllables as same or different. This approach was chosen because it has repeatedly been shown to be a useful data compression step (Schorff et al. 2015, Thakur et al. 2019) before further downstream tasks are applied on its output, particularly in contexts where there is little data per class (syllable label). For example, Schorff et al. 2015 trained a deep convolutional neural network with triplet loss to embed images of human faces from the same individual closer together than images of different individuals in a 128dimensional space. They then used this model to compute 128-dimensional representations of additional face images, not included in training, which were used for individual facial recognition (this is a same vs. different category classifier), and facial clustering, achieving better performance than the previous state of the art. The triplet loss function results in a model that can generate useful embeddings of previously unseen categories, like new individuals’ faces, or new zebra finches’ syllables, which can then be used in downstream analyses. This meaningful, lower dimensional space allows comparisons of distributions of syllables across birds, as in Brainard and Mets 2008, and Goffinet et al. 2021.

Next word and masked word prediction are indeed common self-supervised learning tasks for models working with text data, or other data with meaningful sequential organization. That is not the case for our zebra finch syllables, where every bird’s syllable sequence depends only on its tutor’s sequence, and there is no evidence for strong universal syllable sequencing rules (James et al. 2020). Rather, our embedding model is an example of a computer vision task, as it deals with sets of two-dimensional images (spectrograms), not sequences of categorical variables (like text). It is also not, strictly speaking, a selfsupervised learning task, as it does require syllable labels to generate the triplets. A common selfsupervised approach for dimensionality reduction in a computer vision task such as this one would be to train an autoencoder to compress images to a lower dimensional space, then faithfully reconstruct them from the compressed representation. This has been done using a variational autoencoder trained on zebra finch syllables in Goffinet et al. 2021. In keeping with the suggestions from reviewers #1 and #2, we have included a comparison of our triplet loss model with the Goffinet et al. VAE approach in the revised manuscript.

(2) The machine learning methodology lacks rigor. The aims of the machine learning pipeline are extremely vague and keep changing like a moving target. Mainly, the deep networks are trained on some tasks but then authors evaluate their performance on different, disconnected tasks. For example, they train both the birdsong comparison method (L263+) and the song similarity method (L318+) on classification tasks. However, they evaluate the former method (LDA) on classification accuracy, but the latter (8-dim embeddings) using a contrast index. In machine learning, usually, a useful task is first defined, then the system is trained on it and then tested on a held-out dataset. If the sensitivity index is important, why does it not serve as a cost function for training?

Again, this reviewer seems not to understand our similarity scoring methodology. Our similarity scoring model is not trained on a classification task, but rather on an embedding task. It learns to embed spectrograms of syllables in an 8-dimensional space such that syllables with the same label are closer together than syllables with different labels. We could report the loss values for this embedding task on our training and validation datasets, but these wouldn’t have any clear relevance to the downstream task of syllable distribution comparison where we are using the model’s embeddings. We report the contrast index as this has direct relevance to the actual application of the model and allows comparisons to other similarity scoring methods, something that the triplet loss values wouldn’t allow.

The triplet loss method was chosen because it has been shown to yield useful low-dimensional representations of data, even in cases where there is limited labeled training data (Thakur et al. 2019). While we have one of the largest manually annotated datasets of zebra finch songs, it is still quite small by industry deep learning standards, which is why we chose a method that would perform well given the size of our dataset. Training a model on a contrast index directly would be extremely computationally intensive and require many more pairs of birds with known relationships than we currently have access to. It could be an interesting approach to take in the future, but one that would be unlikely to perform well with a dataset size typical to songbird research.

Also, usually, in solid machine learning work, diverse methods are compared against each other to identify their relative strengths. The paper contains almost none of this, e.g. authors examined only one clustering method (HDBSCAN).

We did compare multiple methods for syllable segmentation (WhisperSeg, TweetyNet, and Amplitude thresholding) as this hadn’t been done previously. We chose not to perform extensive comparison of different clustering methods as Sainburg et al. 2020 already did so and we felt no need to reduplicate this effort. We encourage this reviewer to refer to Sainburg et al.’s excellent work for comparisons of multiple clustering methods applied to zebra finch song syllables.

(3) Performance issues. The authors want to 'simplify large-scale behavioral analysis' but it seems they want to do that at a high cost. (Gu et al 2023) achieved syllable scores above 0.99 for adults, which is much larger than the average score of 0.88 achieved here (L121). Similarly, the syllable scores in (Cohen et al 2022) are above 94% (their error rates are below 6%, albeit in Bengalese finches, not zebra finches), which is also better than here. Why is the performance of AVN so low? The low scores of AVN argue in favor of some human labeling and training on each bird.

Firstly, the syllable error rate scores reported in Cohen et al. 2022 are calculated very differently than the F1 scores we report here and are based on a model trained with data from the same bird as was used in testing, unlike our more general segmentation approach where the model was tested on different birds than were used in training. Thus, the scores reported in Cohen et al. and the F1 scores that we report cannot be compared.

The discrepancy between the F1_seg_ scores reported in Gu et al. 2023 and the segmentation F1 scores that we report are likely due to differences in the underlying datasets. Our UTSW recordings tend to have higher levels of both stationary and non-stationary background noise, which make segmentation more challenging. The recordings from Rockefeller were less contaminated by background noise, and they resulted in slightly higher F1 scores. That said, we believe that the primary factor accounting for this difference in scores with Gu et al. 2023 is the granularity of our ‘ground truth’ syllable segments. In our case, if there was never any ambiguity as to whether vocal elements should be segmented into two short syllables with a very short gap between them or merged into a single longer syllable, we chose to split them. WhisperSeg had a strong tendency to merge the vocal elements in ambiguous cases such as these. This results in a higher rate of false negative syllable onset detections, reflected in the low recall scores achieved by WhisperSeg (see Figure 2–figure supplement 1b), but still very high precision scores (Figure 2–figure supplement 1a). While WhisperSeg did frequently merge these syllables in a way that differed from our ground truth segmentation, it did so consistently, meaning it had little impact on downstream measures of syntax entropy (Figure 3c) or syllable duration entropy (Figure 3–figure supplement 2a). It is for that reason that, despite a lower F1 score, we still consider AVN’s automatically generated annotations to be sufficiently accurate for downstream analyses.

Should researchers require a higher degree of accuracy and precision with their annotations (for example, to detect very subtle changes in song before and after an acute manipulation) we suggest they turn toward one of the existing tools for supervised song annotation, such as TweetyNet.

(4) Texas bias. It is true that comparability across datasets is enhanced when everyone uses the same code. However, the authors' proposal essentially is to replace the bias between labs with a bias towards birds in Texas. The comparison with Rockefeller birds is nice, but it amounts to merely N=1. If birds in Japanese or European labs have evolved different song repertoires, the AVN might not capture the associated song features in these labs well.

We appreciate the author’s concern about a bias toward birds from the UTSW colony. However, this paper shows that despite training (for the similarity scoring) and hyperparameter fitting (for the HDBSCAN clustering) on the UTSW birds, AVN performs as well if not better on birds from Rockefeller than from UTSW. To our knowledge, there are no publicly available datasets of annotated zebra finch songs from labs in Europe or in Asia but we would be happy to validate AVN on such datasets, should they become available. Furthermore, there is no evidence to suggest that there is dramatic drift in zebra finch vocal repertoire between continents which would necessitate such additional validation. While we didn’t have manual annotations for this dataset (which would allow validation of our segmentation and labeling methods), we did apply AVN to recordings shared with us by the Wada lab in Japan, where visual inspection of the resulting annotations suggested comparable accuracy to the UTSW and Rockefeller datasets.

(5) The paper lacks an analysis of the balance between labor requirement, generalizability, and optimal performance. For tasks such as segmentation and labeling, fine-tuning for each new dataset could potentially enhance the model's accuracy and performance without compromising comparability. E.g. How many hours does it take to annotate hundred song motifs? How much would the performance of AVN increase if the network were to be retrained on these? The paper should be written in more neutral terms, letting researchers reach their own conclusions about how much manual labor they want to put into their data.

With standardization and ease of use in mind, we designed AVN specifically to perform fully automated syllable annotation and downstream feature calculations. We believe that we have demonstrated in this manuscript that our fully automated approach is sufficiently reliable for downstream analyses across multiple zebra finch colonies. That said, if researchers require an even higher degree of annotation precision and accuracy, they can turn toward one of the existing methods for supervised song annotation, such as TweetyNet. Incorporating human annotations for each bird processed by AVN is likely to improve its performance, but this would require significant changes to AVN’s methodology, and is outside the scope of our current efforts.

(6) Full automation may not be everyone's wish. For example, given the highly stereotyped zebra finch songs, it is conceivable that some syllables are consistently mis-segmented or misclassified. Researchers may want to be able to correct such errors, which essentially amounts to fine-tuning AVN. Conceivably, researchers may want to retrain a network like the AVN on their own birds, to obtain a more fine-grained discriminative method.

Other methods exist for supervised or human-in-the-loop annotation of zebra finch songs, such as TweetyNet and DAN (Alam et al. 2023). We invite researchers who require a higher degree of accuracy than AVN can provide to explore these alternative approaches for song annotation. Incorporating human feedback into AVN was never the goal of our pipeline, would require significant changes to AVN’s design and is outside the scope of this manuscript.

(7) The analysis is restricted to song syllables and fails to include calls. No rationale is given for the omission of calls. Also, it is not clear how the analysis deals with repeated syllables in a motif, whether they are treated as two-syllable types or one.

It is true that we don’t currently have any dedicated features to describe calls. This could be a useful addition to AVN in the future.

What a human expert inspecting a spectrogram would typically call ‘repeated syllables’ in a bout are almost always assigned the same syllable label by the UMAP+HDBSCAN clustering. The syntax analysis module includes features examining the rate of syllable repetitions across syllable types, as mentioned in lines 222-226 of the revised manuscript. See https://avn.readthedocs.io/en/latest/syntax_analysis_demo.html#Syllable-Repetitions for further details.

(8) It seems not all human annotations have been released and the instruction sets given to experts (how to segment syllables and score songs) are not disclosed. It may well be that the differences in performance between (Gu et al 2023) and (Cohen et al 2022) are due to differences in segmentation tasks, which is why these tasks given to experts need to be clearly spelled out. Also, the downloadable files contain merely labels but no identifier of the expert. The data should be released in such a way that lets other labs adopt their labeling method and cross-check their own labeling accuracy.

All human annotations used in this manuscript have indeed been released as part of the accompanying dataset. Syllable annotations are not provided for all pupils and tutors used to validate the similarity scoring, as annotations are not necessary for similarity comparisons. We have expanded our description of our annotation guidelines in the methods section of the revised manuscript. All the annotations were generated by one of two annotators. The second annotator always consulted with the first annotator in cases of ambiguous syllable segmentation or labeling, to ensure that they had consistent annotation styles. Unfortunately, we haven’t retained records about which birds were annotated by which of the two annotators, so we cannot share this information along with the dataset. The data is currently available in a format that should allow other research groups to use our annotations either to train their own annotation systems or check the performance of their existing systems on our annotations.

(9) The failure modes are not described. What segmentation errors did they encounter, and what syllable classification errors? It is important to describe the errors to be expected when using the method.

As we discussed in our response to this reviewer’s point (3), WhisperSeg has a tendency to merge syllables when the gap between them is very short, which explains its lower recall score compared to its precision on our dataset (Figure 2–figure supplement 1). In rare cases, WhisperSeg also fails to recognize syllables entirely, again impacting its precision score. TweetyNet hardly ever completely ignores syllables, but it does tend to occasionally merge syllables together or over-segment them. Whereas WhisperSeg does this very consistently for the same syllable types within the same bird, TweetyNet merges or splits syllables more inconsistently. This inconsistent merging and splitting has a larger effect on syllable labeling, as manifested in the lower clustering v-measure scores we obtain with TweetyNet compared to WhisperSeg segmentations. TweetyNet also has much lower precision than WhisperSeg, largely because TweetyNet often recognizes background noises (like wing flaps or hopping) as syllables whereas WhisperSeg hardly ever segments non-vocal sounds.

Many errors in syllable labeling stem from differences in syllable segmentation. For example, if two syllables with labels ‘a’ and ‘b’ in the manual annotation are sometimes segmented as two syllables, but sometimes merged into a single syllable, the clustering is likely to find 3 different syllable types; one corresponding to ‘a’, one corresponding to ‘b’ and one corresponding to ‘ab’ merged. Because of how we align syllables across segmentation schemes for the v-measure calculation, this will look like syllable ‘b’ always has a consistent cluster label (or is missing a label entirely), but syllable ‘a’ can carry two different cluster labels, depending on the segmentation. In certain cases, even in the absence of segmentation errors, a group of syllables bearing the same manual annotation label may be split into 2 or 3 clusters (it is extremely rare for a single manual annotation group to be split into more than 3 clusters). In these cases, it is difficult to conclusively say whether the clustering represents an error, or if it actually captured some meaningful systematic difference between syllables that was missed by the annotator. Finally, sometimes rare syllable types with their own distinct labels in the manual annotation are merged into a single cluster. Most labeling errors can be explained by this kind of merging or splitting of groups relative to the manual annotation, not to occasional mis-classifications of one manual label type as another.

For examples of these types of errors, we encourage this reviewer and readers to refer to the example confusion matrices in figure 2f and Figure 2–figure supplement 3b&e. We also added two paragraphs to the end of the ‘Accurate, fully unsupervised syllable labeling’ section of the Results in the revised manuscript.

(10) Usage of Different Dimensionality Reduction Methods: The pipeline uses two different dimensionality reduction techniques for labeling and similarity comparison - both based on the understanding of the distribution of data in lower-dimensional spaces. However, the reasons for choosing different methods for different tasks are not articulated, nor is there a comparison of their efficacy.

We apologize for not making this distinction sufficiently clear in the manuscript and have added a paragraph to the ‘Measuring Song Imitation’ section of the Results explaining the rational for using an embedding model for similarity scoring.

We chose to use UMAP for syllable labeling because it is a common embedding methodology to precede hierarchical clustering and has been shown to result in reliable syllable labels for birdsong in the past (Sainburg et al. 2020). However, it is not appropriate for similarity scoring, because comparing EMD or MMD scores between birds requires that all the birds’ syllable distributions exist within the same shared embedding space. This can be achieved by using the same triplet loss-trained neural network model to embed syllables from all birds. This cannot be achieved with UMAP because all birds whose scores are being compared would need to be embedded in the same UMAP space, as distances between points cannot be compared across UMAPs. In practice, this would mean that every time a new tutor-pupil pair needs to be scored, their syllables would need to be added to a matrix with all previously compared birds’ syllables, a new UMAP would need to be computed, and new EMD or MMD scores between all bird pairs would need to be calculated using their new UMAP embeddings. This is very computationally expensive and quickly becomes unfeasible without dedicated high power computing infrastructure. It also means that similarity scores couldn’t be compared across papers without recomputing everything each time, whereas EMD and MMD scores obtained with triplet loss embeddings can be compared, provided they use the same trained model (which we provide as part of AVN) to embed their syllables in a common latent space.

(11) Reproducibility: are the measurements reproducible? Systems like UMAP always find a new embedding given some fixed input, so the output tends to fluctuate.

There is indeed a stochastic element to UMAP embeddings which will result in different embeddings and therefore different syllable labels across repeated runs with the same input. We observed that v-measures scores were quite consistent within birds across repeated runs of the UMAP, and have added an additional supplementary figure to the revised manuscript showing this (Figure 2–figure supplement 4).

**Reviewer #1 (Recommendations For The Authors):**
(1) Benchmark their similarity score to the method used by Goffinet et al, 2021 from the Pearson group. Such a comparison would be really interesting and useful.

This has been added to the paper.

(2) Please clarify exactly what is new and what is applied from existing methods to help the reader see the novelty of the paper.

We have added more emphasis on the novel aspects of our pipeline to the paper’s introduction.

Minor:It's unclear if AVN is appropriate as the paper deals only with zebra finch song - the scope is more limited than advertised.

We assume this is in reference to ‘Birdsong’ in the paper’s title and ‘Avian’ in Avian Vocalization Network. There is a brief discussion of how these methods are likely to perform on other commonly studied songbird species at the end of the discussion section.

**Reviewer #2 (Recommendations For The Authors):**
A few points for the authors to consider that might strengthen or inform the paper:(1) In the public review, I detailed some ways in which the SSL+EMD approach is unlikely to be appreciably distinct from the VAE+MMD approach -- in fact, one could mix and match here. It would strengthen the authors' claim if they showed via experiments that their method outperforms VAE+MMD, but in the absence of that, a discussion of the relation between the two is probably warranted.

This comparison has been added to the paper.

(2) ll. 305-310: This loss of accuracy near the edge is expected on general Bayesian grounds. Any regression approach should learn to estimate the conditional mean of the age distribution given the data, so ages estimated from data will be pulled inward toward the location of most training data. This bias is somewhat mitigated in the Brudner paper by a more flexible model, but it's a general (and expected) feature of the approach.(3) While the online AVA documentation looks good, it might benefit from a page on design philosophy that lays out how the various modules fit together - something between the tutorials and the nitty-gritty API. That way, users would be able to get a sense of where they should look if they want to harness pieces of functionality beyond the tutorials.

Thank you for this suggestion. We will add a page on AVN’s design philosophy to the online documentation.

(4) While the manuscript does compare AVN to packages like TweetyNet and AVA that share some functionality, it doesn't really mention what's been going on with the vocalpy ecosystem, where the maintainers have been doing a lot to standardize data processing, integrate tools, etc. I would suggest a few words about how AVN might integrate with these efforts.

We thank the reviewer for this suggestion.

(5) ll. 333-336: It would be helpful to provide a citation to some of the self-supervised learning literature this procedure is based on. Some citations are provided in methods, but the general approach is worth citing, in my opinion.

We have added a paragraph to the results section with more background on self-supervised learning for dimensionality reduction, particularly in the context of similarity scoring.

(6) One software concern for medium-term maintenance: AVN docs say to use Python 3.8, and GitHub says the package is 3.9 compatible. I also saw in the toml file that 3.10 and above are not supported. It's worth noting that Python 3.9 reaches its end of life in October 2025, so some dependencies may have to be altered or changed for the package to be viable going forward.

Thank you for this comment. We will continue to maintain AVN and update its dependencies as needed.

Minor points:(1) It might be good to note that WhisperSeg is a different install from AVN. May be hard for novice users, though there's a web interface that's available.

We’ve added a line to the methods section making this clear.

(2) Figure 6b: Some text in the y-axis labels is overlapping here.

This has been fixed. Thank you for bringing it to our attention.

(3) The name of the Python language is always capitalized.

We’ve fixed this capitalization error throughout the manuscript. Thank you.

**Reviewer #3 (Recommendations For The Authors):**
(1) I recommend that the authors improve the motivation of the chosen tasks and data or choose new tasks that more clearly speak to the optimizations they want to perform.

We have included more details about the motivation for our LDA classification analysis, age prediction model and embedding model for similarity scoring in the results of the revised manuscript, as discussed in more detail in the above responses to this reviewer. Thank you for these suggestions.

(2) They need to rigorously report the (classification) scores on the test datasets: these are the scores associated with the cost function used during training.

Based on this reviewer’s ‘Weaknesses: 3’ comment in the public reviews, we believe that they are referring to a classification score for the triplet loss model. As we explained in response to that comment, this is not a classification task, therefor there is no classification score to report. The loss function used to train the model was a triplet loss function. While we could report these values, they are not informative for how well this approach would perform in a similarity scoring context, as explained above. As such, we prefer to include contrast index and tutor contrast index scores to compare the models’ performance for similarity score, as these are directly relevant to the task and are established in the field for said task.

(3) They need to explain the reasons for the poor performance (or report on the inconsistencies with previous work) and why they prefer a fully automated system rather than one that needs some fine-tuning on bird-specific data.

We’ve addressed this comment in the public response to this reviewer’s weakness points 3, 5, and 6.

(4) They should consider applying their method to data from Japanese and European labs.

We’ve addressed this comment in the public response to this reviewer’s weakness point 4.

(5) The need to document the failure modes and report all details about the human annotations.

We’ve added additional description of the failure modes for our segmentation and labeling approaches in the results section of the revised manuscript.

Details:The introduction is very vague, it fails to make a clear case of what the problem is and what the approach is. It reads a bit like an advertisement for machine learning: we are given a hammer and are looking for a nail.

We thank the reviewer for this viewpoint; however, we disagree and have decided to keep our Introduction largely unchanged.

L46 That interpretability is needed to maximize the benefits of machine learning is wrong, see self-driving cars and chat GPT.

This line states that ‘To truly maximize the benefits of machine learning and deep learning methods for behavior analysis, their power must be balanced with interpretability and generalizability’. We firmly believe that interpretability is critically important when using machine learning tools to gain a deeper scientific understanding of data, including animal behavior data in a neuroscience context. We believe that the introduction and discussion of this paper already provide strong evidence for this claim.

L64 What about zebra finches that repeat a syllable in the motif, how are repetitions dealt with by AVN?

This is already described in the results section in lines 222-226, and in the methods in the ‘Syntax Features: Repetition Bouts’ section.

L107 Say a bit more here, what exactly has been annotated?

We’ve added a sentence in the introduction to clarify this. Line 113-115.

L112 Define spectrogram frames. Do these always fully or sometimes partially contain a vocalization?

Spectrogram frames are individual time bins used to compute the spectrogram using a short-term Fourier transform. As described in the ‘Methods; Labeling : UMAP Dimensionality Reduction” section, our spectrograms are computed using ‘The short term Fourier transform of the normalized audio for each syllable […] with a window length of 512 samples and a hop length of 128 samples’. Given that the song files have a standard sampling rate of 44.1kHz, this means each time bin represents 11.6ms of song data, with successive frames advancing in time by 2.9ms. These contain only a small fraction of a vocalization.

L122 The reported TweetyNet score of 0.824 is lower than the one reported in Figure 2a.

The center line in the box plot in Figure 2a represents the median of the distribution of TweetyNet vmeasure scores. Given that there are a couple outlying birds with very low scores, the mean (0.824 as reported in the text of the results section) is lower than the median. This is not an error.

L155 Some of the differences in performance are very small, reporting of the P value might be necessary.

These methods are unlikely to statistically significantly differ in their validation scores. This doesn’t mean that we cannot use the mean/median values reported to justify favoring one method over another. This is why we’ve chosen not to report p-values here.

L161 The authors have not really tested more than a single clustering method, failing to show a serious attempt to achieve good performance.

We’ve addressed this comment in the public response to this reviewer’s weakness point 2.

L186 Did isolate birds produce stereotyped syllables that can be clustered?

Yes, they did. The validation for clustering of isolate bird songs can be found in Figure 2–figure supplement 4.

Fig. 3e: How were the multiple bouts aligned?

This is described in lines 857-876 in the ‘Methods: Song Timing Features: Rhythm Spectrograms” section of the paper.

L199 There is a space missing in front of (n=8).

Thank you for bringing this to our attention. It’s been corrected in the updated manuscript.

L268 Define classification accuracy.

We’ve added a sentence in lines 953-954 of the methods section defining classification accuracy.

L325 How many motifs need to be identified, why does this need to be done manually? There are semiautomated methods that can allow scaling, these should be cited here. Also, the mention of bias here should be removed in favor of a more extensive discussion on the experimenter bias (traditionally vs Texas bias in this paper).

All of the methods cited in this line have graphical user interfaces that require users to select a file containing song and manually highlight the start and end each motif to be compared. The exact number of motifs required varies depending on the specific context (e.g. more examples are needed to detect more subtle differences or changes in song similarity) but it is fairly standard for reviewers to score 30 – 100 pairs of motifs.

We’ve discussed the tradeoffs between full automation and supervised or human-in-the loop methods in response to this reviewer’s public comment ‘weakness #5 and 6’. Briefly, AVN’s aim is to standardize song analysis, to allow direct comparisons between song features and similarity scores across research groups. We believe, as explained in the paper, that this can be best achieve by having different research groups use the same deep learning models, which perform consistently well across those groups. Introducing semi-automated methods would defeat this benefit of AVN.

We’ve also addressed the question of ‘Texas bias’ in response to their reviewer’s public comment ‘Weakness #4’.

L340 How is EMD applied? Syllables are points in 8-dim space, but now suddenly authors talk about distributions without explaining how they got from points to distributions. Same in L925.

We apologize for the confusion here. The syllable points in the 8-d space are collectively an empirical distribution, not a probability distribution. We referred to them simply as ‘distributions’ to limit technical jargon in the results of the paper, but have changed this to more precise language in the revised manuscript.

L351 Why do authors now use 'contrast index' to measure performance and no longer 'classification accuracy'?

We’ve addressed this comment in the public response to this reviewer’s weakness points 1 and 2.

Figure 6 What is the confusion matrix, i.e. how well can the model identify pupil-pupil pairings from pupiltutor and from pupil-unrelated pairings? I guess that would amount to something like classification accuracy.

There is no model classifying comparisons as pupil-pupil vs. pupil-tutor etc. These comparisons exist only to show the behavior of the similarity scoring approach, which consists of a dissimilarity measure (MMD or EMD) applied to low dimensional representations of syllable generated by the triplet loss model or VAE. This was clarified further in our public response to this reviewer’s weakness points 1 and 2.

L487 What are 'song files', and what do they contain?

‘Song files’ are .wav files containing recordings of zebra finch song. They typically contain a single song bout, but they can include multiple song bouts if they are produced close together, or incomplete song bouts if the introductory notes were very soft or the bouts were very long (>30s from the start of the file). Details of these recordings are provided in the ‘Methods: Data Acquisition: UTSW Dataset’ section of the manuscript.

L497 Calls were only labelled for tweetynet but not for other tasks.

That is correct. The rationale for this is provided in the ‘Methods: Manual Song Annotation’ section of the manuscript.

L637 There is a contradiction (can something be assigned to the 'own manual annotation category' when the same sentence states that this is done 'without manual annotation'?)

We believe there is confusion here between automated annotation and validation. Any bird can be automatically annotated without the need for any existing manual annotations for that individual bird. However, manual labels are required to compare automatically generated annotations against for validation of the method.

L970 Spectograms of what? (what is the beginning of a song bout, L972).

The beginning of a song bout is the first introductory note produced by a bird after a period without vocalizations. This is standard.